# Towards Robustness and Explainability of Automatic Algorithm Selection

Xingyu Wu [1]   Jibin Wu [1 2]   Yu Zhou [1]   Liang Feng [3]   Kay Chen Tan [1]

## Abstract

Algorithm selection aims to identify the optimal performing algorithm before execution. Existing techniques typically focus on the observed correlations between algorithm performance and meta-features. However, little research has explored the underlying mechanisms of algorithm selection, specifically what characteristics an algorithm must possess to effectively tackle problems with certain feature values. This gap not only limits the explainability but also makes existing models vulnerable to data bias and distribution shift. This paper introduces directed acyclic graph (DAG) to describe this mechanism, proposing a novel modeling paradigm that aligns more closely with the fundamental logic of algorithm selection. By leveraging DAG to characterize the algorithm feature distribution conditioned on problem features, our approach enhances robustness against marginal distribution changes and allows for finer-grained predictions through the reconstruction of optimal algorithm features, with the final decision relying on differences between reconstructed and rejected algorithm features. Furthermore, we demonstrate that, the learned graph and the proposed counterfactual calculations offer our approach with both model-level and instance-level explainability.

## 1. Introduction

With the advancement of information technology, a multitude of algorithms has emerged across various fields, making automated algorithm selection increasingly vital. This necessity arises from the fact that a single algorithm often struggles to perform well across all scenarios (Tornede et al., 2023; Pio et al., 2024). The goal of automated algorithm selection is to identify the most suitable algorithm for each specific problem instance, thereby enhancing performance in terms of accuracy (Ruhkopf et al., 2023) and time efficiency (Kostovska et al., 2023). Early methods focused on identifying suitable meta-features to characterize problems, such as statistical characteristics (Hutter et al., 2014) and probing features (Nudelman et al., 2004). These approaches employed machine learning techniques to meta-learn which algorithm performs best for a given problem instance (Abdulrahman et al., 2018). More recent studies have also recognized the importance of algorithm meta-features, such as hyperparameters (Tornede et al., 2020) and code-related features (Pulatov et al., 2022; Wu et al., 2024). Details of existing techniques are provided in **Appendix A.1**.

However, regardless of whether algorithm features are included in the meta-features, existing methods predominantly construct empirical models based on observed correlations between algorithm performance and meta-features (Kerschke et al., 2019; Xu et al., 2008). These approaches overlook the underlying mechanism that drives the algorithm selection task, i.e.: **What characteristics an algorithm needs to solve a problem with specific feature values.** By exploring this mechanism, we can align more closely with the foundational logic of algorithm selection, moving beyond mere predictions of the optimal algorithm or algorithm performance based on meta-features. This granular modeling approach not only facilitates a more effective utilization of algorithm features but also allows for deeper insights into model decisions at the feature level, rather than solely from the perspective of performance comparison.

Moreover, the core of this mechanism lies in modeling the conditional distribution of algorithm features based on problem features, rather than relying on the marginal distribution of meta-features for optimal algorithm prediction. This shift can address a significant challenge in current techniques: their vulnerability to changes in marginal distribution. This issue is prevalent in all models during deployment, as the marginal distribution in algorithm selection is inherently dynamic (Van Rijn et al., 2014; 2018). As illustrated in Figure 1(a), ongoing technological advancements continuously create new application domains and candidate algorithms, leading to fluctuating strengths and weaknesses among them.

---
[1]Department of Data Science and Artificial Intelligence, The Hong Kong Polytechnic University, Hong Kong SAR, China [2]Department of Computing, The Hong Kong Polytechnic University, Hong Kong SAR, China [3]College of Computer Science, Chongqing University, Chongqing, China. Correspondence to: Jibin Wu <jibin.wu@polyu.edu.hk>.

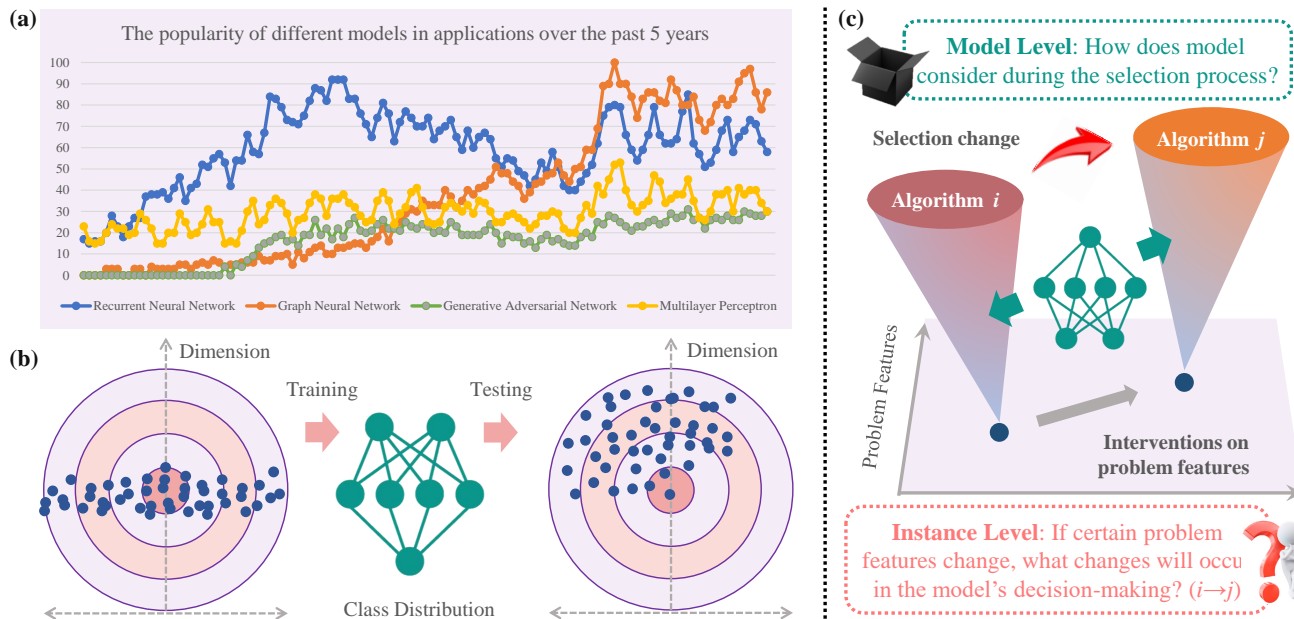

Figure 1. Illustration of the impact of neglecting underlying mechanisms: (a) Shifts in the popularity of classical algorithms over the past decade, highlighting data bias in algorithm selection tasks; (b) An example of problem distribution shift, where meta-training data are small-scale problems collected from uniform distribution, while the test problems are high-dimensional with imbalanced distribution; (c) Two levels of the lacking explainability-unveiling the decision-making mechanisms of the "black box" models, and providing counterfactual explanations for the selection outcomes.

Consequently, the meta-training data collected at any point may not accurately reflect the evolving distribution of problems and algorithms. Furthermore, discrepancies between training and deployment environments are widespread (Wu et al., 2024). For instance, as shown in Figure 1(b), the meta-training data may come from relatively small-scale, uniformly distributed problems, while the deployment environment could involve high-dimensional and imbalanced problems. Additionally, meta-training data is often skewed towards mainstream algorithms, resulting in inadequate representation for lesser-known or newly proposed alternatives.

This paper introduces causality to describe this underlying mechanism, taking a significant step towards robustness and explainability. Herein, the causal relationships imply that: *Because* certain problem features are present, the most suitable algorithm must exhibit specific characteristics to effectively address the problem. Under reasonable assumptions, we employ the structural equation model (Bongers et al., 2021) to construct a directed acyclic graph (DAG) that captures the conditional distribution in the algorithm selection task. Hence, the identified variable relations remain robust against both covariate and label shifts (Bühlmann, 2020; Oberst et al., 2021), thereby mitigating the impact of changes in problem and algorithm distributions. Inspired by the gradient-based causal learning and recommendation (Zheng et al., 2018; He et al., 2022), deep neural networks are employed to identify the reconstruction mechanism in

meta-training data, quantifying the strength of causality among variables. The reconstructed algorithm features can represent the feature values of the most suitable algorithm for solving a problem instance. Our framework ranks candidate algorithms by comparing the reconstructed features of the selected algorithm with those of the rejected ones, leading to a final selection. During training, the model optimization is guided by a combination of reconstruction error, acyclicity constraints, sparsity penalty, and ranking error.

Beside the performance superiority, causal modeling paradigm offers multi-level explainability for algorithm selection. First, the construction of the DAG visually illustrates the interdependencies among variables, while the structural equation model (Pearl, 2012) provides feature-level predictions for the "most suitable algorithm," clearly demonstrating the comprehensive mechanism underlying the model decision. Furthermore, counterfactual explanations (Guidotti, 2022) for each individual problem instance can be achieved by manipulating problem features within the structural equations, as illustrated in Figure 1(c). The goal of manipulation is to identify the minimal interventions required to alter the algorithm selection outcome for each problem instance, clarifying which feature values make the selected algorithm most suitable for solving this problem instance. Following the principle of Occam's Razor (Rasmussen & Ghahramani, 2000), we measure the minimal intervention from the perspectives of explanation complex-

ity and explanation strength, where complexity pertains to the number of modified features, while strength relates to the magnitude of the changes induced by the intervention. By solving this optimization problem, we can ultimately derive the counterfactual explanations for each problem instance.

In this paper, we propose the DAG-based algorithm selection (DAG-AS)[1]. Our key contributions are summarized as:

- **Robustness**: DAG-AS models the invariant causal mechanisms underlying algorithm selection process, which not only enables more accurate selection decisions, but also enhances robustness against common forms of data bias and distribution shifts that may exist across the problem and algorithm sets.

- **Explainability**: Centered around the learned graph, DAG-AS provides multi-level explainability. At the model level, it effectively unveils the "black box" by elucidating variable relationships through the causal graph and detailing the computational logic via structural equations. At the instance level, DAG-AS achieves counterfactual explanations that reveal how specific feature values influence the selection decision.

- We demonstrate the superiority of DAG-AS in terms of accuracy, robustness, and explainability using the ASlib benchmark. Our analysis of the causal graph underscored the importance of considering algorithm features and causal mechanisms.

## 2. DAG-based Algorithm Selection

We introduce the unified notation used in this paper to discuss the causal relationships among these features within the framework of causal graphs (Pearl, 2016). We employ the calligraphic notation $\mathbb{P}$ to represent the joint probability distribution, and the script notation $\mathcal{G} = \{\mathcal{V}, \mathcal{E}\}$ to denote the DAG, where $\mathcal{V}$ and $\mathcal{E}$ represent the node set and edge set of $\mathcal{G}$, respectively. The adjacency matrix of $\mathcal{G}$ is denoted as $\mathcal{M}$, where $\mathcal{M}_{ij} = 1$ indicates the existence of an edge from the $i$-th node to the $j$-th node, i.e., $(i, j) \in \mathcal{E}$, and 0 otherwise. The definitions of all causal learning-related terms involved in this paper can be found in **Appendix A.2**. Algorithm selection task also involves different feature types. We use $\mathcal{P}$ and $\mathcal{A}$ to represent the problem set and algorithm set, respectively. The bold notations **PF** and **AF** are employed to denote the problem feature set and algorithm feature set, with $\mathcal{V} = \mathbf{PF} \cup \mathbf{AF}$. Specifically, $\mathbf{PF}_i(p)$ represents the $i$-th feature of the $p$-th problem, and $\mathbf{AF}_j(a)$ represents the $j$-th feature of the $a$-th algorithm. Furthermore, when describing the model, we use $W_i$ and $\sigma_i$ to denote the parameter matrix and activation function of the $i$-th layer, respectively.

---

[1]The implementation of DAG-AS is available at https://github.com/wuxingyu-ai/DAG-AS.

### 2.1. Analysis of Distribution in Algorithm Selection

This subsection explores the viability of employing causal models in the algorithm selection task. We will first analyze the alignment between two distinct modeling approaches within the context of algorithm selection. Subsequently, we present Theorem 2.4, which demonstrates that under specific assumptions, utilizing a structural causal model enables accurate modeling of the conditional probability distribution for each algorithm feature. Moreover, this causal framework facilitates the feasible prediction of the optimal algorithm feature given the problem features.

In the realm of algorithm selection, the most widely adopted strategy relies on problem feature-based methods, where the distribution learned is denoted as $\mathbb{P}(\mathcal{A}|\mathbf{PF})$ (Kerschke et al., 2019), representing the mapping from problem feature space to candidate algorithm set. On the other hand, some studies have incorporated algorithm features into the selection process, typically learning the distribution $\mathbb{P}(\mathcal{A}|\mathbf{PF}, \mathbf{AF})$ (Pulatov et al., 2022) or $\mathbb{P}(S|\mathbf{PF}, \mathbf{AF})$ (Wu et al., 2024), where $S$ is a binary variable indicating whether the algorithm associated with **AF** should be selected for the problem associated with **PF**. The relationship between $\mathbb{P}(S = 1|\mathbf{PF}, \mathbf{AF})$ and $\mathbb{P}(\mathbf{AF}|\mathbf{PF}, S = 1)$ can be described according to the Bayes' Theorem:

$$\mathbb{P}(\mathbf{AF}|\mathbf{PF}, S = 1) = \frac{\mathbb{P}(S = 1|\mathbf{PF}, \mathbf{AF})\mathbb{P}(\mathbf{AF}|\mathbf{PF})}{\mathbb{P}(S = 1|\mathbf{PF})} \quad (1)$$

In the context of algorithm selection, $\mathbb{P}(\mathbf{AF}|\mathbf{PF})$ represents the likelihood of each algorithm being considered for each problem and is assumed to follow a uniform distribution. $\mathbb{P}(S = 1|\mathbf{PF})$ is also a constant. Therefore, we can attempt to model $\mathbb{P}(\mathbf{AF}|\mathbf{PF}, S = 1)$ as a substitute for $\mathbb{P}(S = 1|\mathbf{PF}, \mathbf{AF})$. Although these two distributions carry similar information, their dependency on the marginal distribution differs from a modeling perspective. $\mathbb{P}(S = 1|\mathbf{PF}, \mathbf{AF})$ relies on the joint distribution $\mathbb{P}(\mathbf{PF}, \mathbf{AF})$, where the model's input space includes both problem and algorithm features. Thus, both covariate and label shifts can impact the model's performance (Park et al., 2023). In contrast, $\mathbb{P}(\mathbf{AF}|\mathbf{PF}, S = 1)$ directly models the conditional distribution, aligning with the underlying mechanism advocated in this paper. As long as this underlying mechanism remains unchanged, the model can maintain resilience to variations in the marginal probability distributions $\mathbb{P}(\mathbf{PF})$ or $\mathbb{P}(\mathbf{AF})$. To model $\mathbb{P}(\mathbf{AF}|\mathbf{PF}, S = 1)$, we first propose three assumptions for the DAG $\mathcal{G}$ in algorithm selection tasks:

**Assumption 2.1.** There exists a DAG $\mathcal{G} = \{\mathcal{V}, \mathcal{E}\}$ on $\mathcal{V} \subset \mathbf{PF} \cup \mathbf{AF}$, and for any variable $\mathbf{AF}_i \in \mathbf{AF}$, the in-degree of $\mathbf{AF}_i$ satisfies $\deg^-(\mathbf{AF}_i) > 0$.

**Assumption 2.2.** For any variable $\mathbf{AF}_i \in \mathbf{AF}$, $\mathbf{Ch}(\mathbf{AF}_i) \cap \mathbf{PF} = \varnothing$.

**Assumption 2.3.** Informative algorithm features **AF** are available for the algorithm selection task.

The first two assumptions are primarily constructed based on the distinctive characteristics of the algorithm selection task, ensuring the directional flow of model computations and simplifying the model structure. Specifically, Assumption 2.1 guarantees that each algorithm feature is predictable, with no exogenous variables present. This is because the prediction of exogenous variables relies on external environmental factors, which are only influenced by noise terms in a structural causal model. Reconstructing algorithm features based on these noise terms would introduce uncertainties into the algorithm selection model. Assumption 2.2 primarily restricts the directionality of causal links within the DAG $\mathcal{G}$, ensuring that the computations in the model flow from problem features to algorithm features. This alignment reflects the core requirement of the algorithm selection, which is to determine the most suitable algorithm based on the given problem features. The few ignored directed edges from algorithm features to problem features do not affect the performance of the proposed model, as we are not concerned with the reconstruction accuracy of problem features. Assumption 2.3 implies that we can obtain algorithm-related characteristics that carry meaningful information for differentiating the performance of different algorithms on various problem instances. This assumption serves as a fundamental basis for DAG-AS. For specific details on how to obtain algorithm features, please refer to **Appendix A.1**.

Grounded in these assumptions, we present Theorem 2.4, which discusses the feasibility of employing causal models to solve the algorithm selection task.

**Theorem 2.4.** *Assume that the joint density function of $\mathbb{P}(\boldsymbol{PF}, \boldsymbol{AF})$ is continuous and strictly positive on a compact and convex subset of $\mathbb{R}^{\boldsymbol{PF} \cup \boldsymbol{AF}}$, and zero elsewhere. And there exists a DAG $\mathcal{G}$ such that $\mathbb{P}(\boldsymbol{PF}, \boldsymbol{AF}|S = 1)$ can be factorized as $\mathbb{P}(\mathcal{V}) = \prod_{V_i \in \mathcal{V}} \mathbb{P}(V_i \mid \boldsymbol{Pa}(V_i))$ according to $\mathcal{G}$. Under Assumption 2.1, any variable $AF_i \in \boldsymbol{AF}$ is not an exogenous variable in $\mathcal{G}$, and there exists a continuous function set $f = \{f_1, f_2, \cdots, f_{|\boldsymbol{AF}|}\}$ with compact support in $\mathbb{R}^{|\boldsymbol{PF}| \times [0,1]}$ such that each $\mathbb{P}(AF_i|\boldsymbol{Pa}(AF_i), S = 1)$ equals the corresponding $f_i$ with uniform distributed noise on $[0, 1]$.*

*Proof.* Please refer to **Appendix B.** $\qquad \square$

Theorem 2.4 establishes that if the underlying distribution can be modeled as a DAG, then each algorithm feature can be associated with a function that accurately captures its conditional probability distribution. Furthermore, this function serves as a mapping from problem features to algorithm features. By modeling the causal relationships, the proposed model can learn the optimal feature values for the most proper algorithm given the problem features, which is the core premise of our approach. We now proceed to construct

a model that can learn the DAG and enable the prediction of the optimal algorithm features for a given problem instance.

## 2.2. Model Framework of DAG-AS

Our proposed model builds upon a neural network framework that incorporates causal learning principles to reconstruct features. The core of our model is the Causal Learning Network, which aims to learn the causal dependencies between variables, including problem features **PF** and algorithm features **AF**. According to Definition 1 in Appendix A, their joint probability distribution over the features can be factorized to $|\mathbf{PF} \cup \mathbf{AF}|$ distinct conditional distributions, which are then modeled by their parent variables and noise terms to account for observed and unobserved factors, respectively. Specifically, the generative process for the problem feature $\mathbf{PF}_i$ and algorithm feature $\mathbf{AF}_i$ can be described as:

$$\mathbf{PF}_i = f_i(\mathbf{Pa}(\mathbf{PF}_i)) + \epsilon_i^{\mathbf{PF}}, \mathbf{Pa}(\mathbf{PF}_i) \subseteq \mathbf{PF} \quad (2)$$

$$\mathbf{AF}_i = g_i(\mathbf{Pa}(\mathbf{AF}_i)) + \epsilon_i^{\mathbf{AF}}, \mathbf{Pa}(\mathbf{AF}_i) \subseteq \mathbf{PF} \cup \mathbf{AF} \quad (3)$$

where $f_i$ and $g_i$ are nonlinear functions representing the causal mechanisms, and $\epsilon_i^{\mathbf{PF}}$ and $\epsilon_i^{\mathbf{AF}}$ are noise terms capturing the unobserved variables. The noise terms are assumed to follow a Uniform distribution.

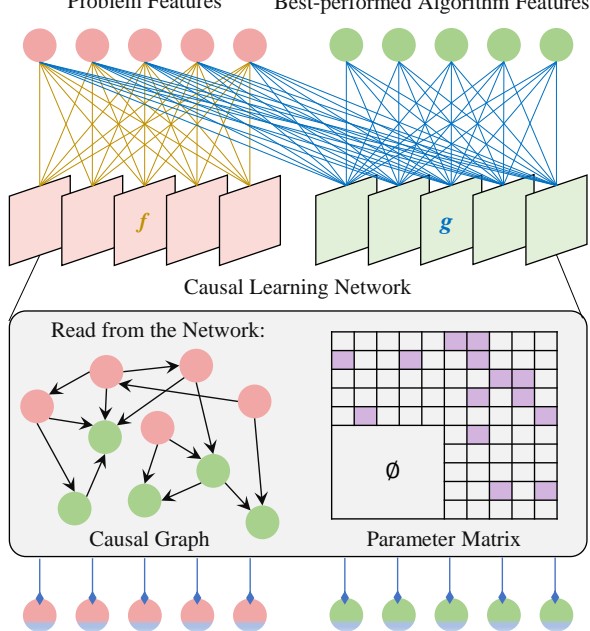

*Figure 2.* Causal learning module for algorithm selection.

As shown in Figure 2, the functions $f$ and $g$ in Eq. (2) and Eq. (3) are denoted using the colors gold and blue, respectively. The input layer comprises nodes representing the features of problem and its best-performed algorithm.

To accurately capture the complex, nonlinear causal relationships between the features, we employ a multi-layer neural network. This network allows for the modeling of intricate dependencies that simpler models might miss. The $l$-th layers, associated with weight matrices $W^{(l)}$ and activation functions $\sigma^{(l)}$, transform the inputs through a series of nonlinear operations. The problem feature reconstruction network ($f$) and the algorithm feature reconstruction network ($g$) are defined as follows:

$$
\begin{aligned}
\hat{\mathbf{PF}}_i &= f(\mathbf{PF}) \leftarrow \\
&W_{L_f}^{(i)}\sigma_{L_f-1}^{(i)}\left(\ldots\sigma_1^{(i)}\left(W_1^{(i)}[\mathbf{PF};\epsilon_i^{\mathbf{PF}}]\right)\right), \\
\hat{\mathbf{AF}}_i &= g(\mathbf{PF},\mathbf{AF}) \leftarrow \\
&W_{L_g}^{(i)}\sigma_{L_g-1}^{(i)}\left(\ldots\sigma_1^{(i)}\left(W_1^{(i)}[\mathbf{PF};\mathbf{AF};\epsilon_i^{\mathbf{AF}}]\right)\right),
\end{aligned} \tag{4}
$$

where $L_f$ and $L_g$ denotes the number of layers in the problem and algorithm feature reconstruction networks, and $[*]$ denotes the concatenation of problem features, algorithm features, and noise.

The adjacency matrix $\mathcal{M}$ of the DAG, which represents the causal structure among the features, is determined through the weight parameters of the first layer of the neural network. Specifically, the non-zero entries in the weight matrix $W_1$ of the first layer indicate the presence of directed edges between nodes:

$$
\mathcal{M}_{ij} = \begin{cases} 1 & \text{if } W_1^{(j)}{}_i \neq 0, \\ 0 & \text{if } W_1^{(j)}{}_i = 0. \end{cases} \tag{5}
$$

where $W_1^{(j)}{}_i$ is the $i$-the element of $W_1^{(j)}$, corresponding to the network parameter of $j$-th feature. This method ensures that the learned causal structure directly influences the parameterization of the neural network, leading to a consistent and interpretable model of the underlying data.

Based on the causal learning module, $\hat{\mathbf{AF}}$ represents the ideal feature values that the best-performing algorithm should have for a given problem. As shown in Figure 3, after obtaining the reconstructed algorithm features, $\hat{\mathbf{AF}}$ and the original algorithm features $\mathbf{AF}$ are each passed through a multi-layer neural network to generate the problem and algorithm representation vectors,

$$
\mathbf{R}_{\text{PF}} = h_{\text{PF}}(\hat{\mathbf{AF}}) \leftarrow W_{L_{\text{PF}}}\sigma_{L_{\text{PF}}}\left(\ldots\sigma_1\left(W_1\hat{\mathbf{AF}}\right)\right), \tag{6}
$$

$$
\mathbf{R}_{\text{AF}} = h_{\text{AF}}(\mathbf{AF}) \leftarrow W_{L_{\text{AF}}}\sigma_{L_{\text{AF}}}\left(\ldots\sigma_1\left(W_1\mathbf{AF}\right)\right), \tag{7}
$$

The similarity between the problem representation $\mathbf{R}_{\text{PF}}$ and the algorithm representation $\mathbf{R}_{\text{AF}}$ is computed through another multi-layer neural network:

$$
s(\mathbf{R}_{\text{PF}},\mathbf{R}_{\text{AF}}) = \text{NN}_{\text{sim}}([\mathbf{R}_{\text{PF}};\mathbf{R}_{\text{AF}}]), \tag{8}
$$

where $\text{NN}_{\text{sim}}$ is the Similarity Computation Module used to calculate the similarity score, and $[\mathbf{R}_{\text{PF}};\mathbf{R}_{\text{AF}}]$ denotes the concatenation of the problem and algorithm representations.

Inspired by (He et al., 2022), the loss function of our model comprises causal learning loss and algorithm selection loss. The causal learning loss includes three components: reconstruction loss, sparsity loss, and acyclicity loss (Zheng et al., 2018). The reconstruction loss ensures that the reconstructed problem and algorithm features $[\hat{\mathbf{PF}},\hat{\mathbf{AF}}]$ are as close as possible to the original features $[\mathbf{PF},\mathbf{AF}]$. This loss captures the fidelity of the causal learning module and ensures that the reconstructed features maintain the essential characteristics of the original features. Mathematically,

$$
\mathcal{L}_{\text{reconstruction}} = \|[\mathbf{PF},\mathbf{AF}] - [\hat{\mathbf{PF}},\hat{\mathbf{AF}}]\|_F^2, \tag{9}
$$

where $\|\cdot\|_F^2$ denotes the Frobenius norm. This term penalizes the squared difference between the original and reconstructed features, encouraging the model to accurately capture the underlying relationships. By minimizing this loss, the model learns to generate algorithm features that are representative of the best algorithm for a given problem. The sparsity loss encourages the adjacency matrix $\mathcal{M}$ of the DAG to be sparse. In a causal graph, sparsity is often desirable as it implies fewer direct dependencies between features, making the graph more interpretable and reducing the risk of overfitting. The sparsity loss is defined as:

$$
\mathcal{L}_{\text{sparsity}} = \|\mathcal{M}\|_0, \tag{10}
$$

where $\|\mathcal{M}\|_0$ denotes the number of non-zero elements in the adjacency matrix. To address the non-convex and non-differentiable nature of the $\ell_0$ norm, in practical deployment one can use the $\|\mathcal{M}\|_1$ norm as an approximation. The acyclicity loss ensures that the learned adjacency matrix $\mathcal{M}$ represents a DAG, which is a fundamental requirement for causal graphs. Acyclicity guarantees that there are no cyclic dependencies among the features, which is crucial for maintaining the logical consistency of causal learning and defined as:

$$
\mathcal{L}_{\text{acyclicity}} = \text{tr}(e^{\mathcal{M}\odot\mathcal{M}}) - |\mathbf{PF}\cup\mathbf{AF}|, \tag{11}
$$

where $\text{tr}(\cdot)$ denotes the trace of a matrix, and $\mathcal{M}\odot\mathcal{M}$ is the Hadamard product (element-wise product) of the adjacency matrix with itself. The algorithm selection loss focuses on optimizing the ranking of algorithms for a given problem. We employ the Bayesian Personalized Ranking loss (Rendle et al., 2009) defined as:

$$
\begin{aligned}
\mathcal{L}_{\text{selection}} = -\sum_{(p,a^+,a^-)} \ln\sigma\big[&s(\mathbf{R}_{\text{PF}}(p),\mathbf{R}_{\text{AF}}(a^+))- \\
&s(\mathbf{R}_{\text{PF}}(p),\mathbf{R}_{\text{AF}}(a^-))\big],
\end{aligned} \tag{12}
$$

where $s(\cdot,\cdot)$ is the similarity score between the problem representation $\mathbf{R}_{\text{PF}}(p)$ and the algorithm representation $\mathbf{R}_{\text{AF}}(a)$,

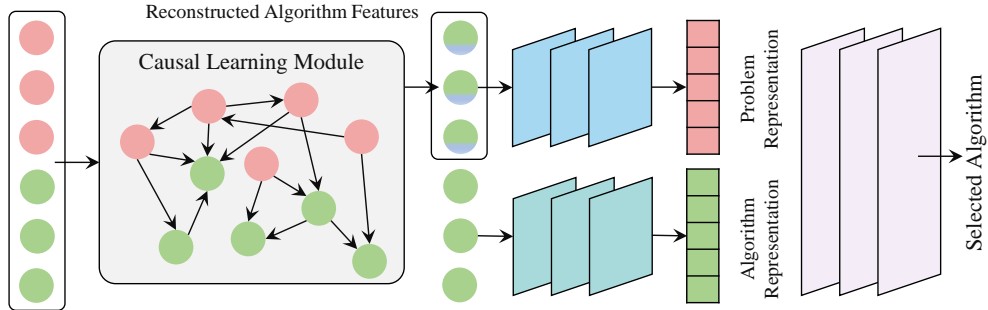

*Figure 3.* Framework of DAG-based algorithm selection.

$a^+$ is the best-performing algorithm, $a^-$ is one of the rejected algorithm, and $\sigma(\cdot)$ is the sigmoid function. The BPR loss works by comparing the similarity scores of the best algorithm $a^+$ and a randomly selected other algorithm $a^-$ for the same problem $p$. By maximizing the difference in similarity scores in favor of $a^+$, the model learns to rank the best algorithm higher.

The overall loss function is the weighted average of the causal learning loss and algorithm selection loss:

$$\mathcal{L} = \alpha\mathcal{L}_{\text{reconstruction}} + \beta\mathcal{L}_{\text{sparsity}} + \gamma\mathcal{L}_{\text{acyclicity}} + \delta\mathcal{L}_{\text{selection}},$$
(13)

where $\alpha$, $\beta$, $\gamma$, and $\delta$ are hyperparameters. In DAG-AS, the reconstruction of problem features is optional. If the focus is solely on algorithm selection results, it suffices to ensure the reconstruction of algorithm features. However, if a more accurate global causal graph is required, the reconstruction of both types of features is essential. The learned DAG will be recorded as an intermediate result to generate explanations in Section 3.1.

### 2.3. Discussion on Applicability of DAG-AS

Based on the introduction of the DAG-AS method, this subsection aims to promptly discuss the applicable scenarios of DAG-AS. The key requirement for applying DAG-AS is the availability of the general configuration for the AS task, encompassing problem features, algorithm features, and performance data. In datasets that possess a rich set of informative features highly relevant to the algorithm's performance, DAG-AS is better positioned to capture the causal relationships. Nevertheless, like any method, DAG-AS has its weaknesses. We believe that the following three scenarios may not be suitable for DAG-AS: (1) When the features themselves lack informativeness or when the causality between problem features and algorithm features is extremely weak. (2) When the data violates the causal sufficiency assumption. (3) When the causal relationships within the dataset are overly complex.

It should be noted, however, that cases (2) and (3) can po-

tentially be rectified by enhancing DAG-AS. In the field of causal learning, there are numerous specialized models crafted to handle complex causal relations, such as those dealing with multivariable or nonlinear causality (Wu et al., 2019; 2022). Moreover, there are studies centered on identifying causality in scenarios where the causal sufficiency (Wu et al., 2023) assumption is violated. To boost the performance of DAG-AS, more advanced causal learning models can be employed to substitute Eq.(6) and (7).

## 3. Explainability of DAG-AS

### 3.1. Model-level Explanation

The model-level explainability is achieved through a detailed analysis of the DAG, which provides insights into how various problem features influence algorithm selection decisions, and enables us to understand the model's inner workings and the rationale behind its algorithm choices. We can dissect this explainability into three key aspects: visualizing relationships, assessing feature importance, and analyzing algorithm-specific influences.

Firstly, the learned graph allows for a visualization of the relationships within all features. In this context, the adjacency matrix $\mathcal{M}$ represents the directed edges between nodes, where $\mathcal{M}_{ij} = 1$ indicates a directed edge from nodes $i$ to $j$. This matrix enables us to visualize how problem features **PF** and algorithm features **AF** are interconnected. Secondly, the structure of the causal graph reveals the in-degree and out-degree of each feature, which indicate their importance in the decision-making process. For example, problem features with zero out-degrees or algorithm features with zero in-degrees, can be considered redundant or irrelevant, demonstrating the feature selection capability inherent in the causal graph. Thirdly, the causal graph is closely related to the final performance of algorithm selection. By examining specific properties exhibited by the DAG, we can explain or estimate the predictive and generalization capabilities of models based on the graph. For instance, when certain causal relationships contradict established domain knowl-

edge or when the causal graph fails to adequately represent variable relationships, the model's performance may deteriorate accordingly. Additionally, an excessively dense causal graph may negatively impact the model's ability to generalize. In the experimental section, we will demonstrate how the causal graph aids us in achieving these explanations.

### 3.2. Instance-level Explanation

This subsection explores the counterfactual explainability of DAG-AS by leveraging interventions (Pearl, 2010) on input features, which allows us to understand the influence from problem features to algorithm features during the algorithm selection process. We introduce perturbations $\delta_{\mathbf{PF}}$ to problem features. To formalize our approach, the Do-Calculus (Huang & Valtorta, 2006), denoted as $do(\cdot)$, is used to represent interventions. The intervention on problem features can be represented as: $\mathbb{P}(S = 1 \mid do(\mathbf{PF} = \mathbf{p} + \delta_{\mathbf{PF}}), \mathbf{AF})$, indicating the probability that $\mathbf{AF}$ is the optimal-performance algorithm features under the intervention $do(\mathbf{PF} = \mathbf{p} + \delta_{\mathbf{PF}})$, where $\mathbf{p}$ denotes the values of $\mathbf{PF}$ before intervention. The goal is to find the minimal perturbations $\delta_{\mathbf{PF}}$ that satisfy some interventional constraint denoted as $h(\mathbb{P}(S = 1 \mid \mathbf{PF}, \mathbf{AF}), \mathbb{P}(S = 1 \mid do(\mathbf{PF} = \mathbf{p} + \delta_{\mathbf{PF}}), \mathbf{AF}))$.

This approach allows us to understand the causal effect of changes in problem features on the algorithm selection decision. Given the DAG-AS model $f(\mathbf{PF}, \mathbf{AF})$, the distribution before and after intervention can be observed in the model output, i.e., the previous output $f(\mathbf{PF}, \mathbf{AF})$ and the counterfactual output $f_\delta = f(\mathbf{PF} + \delta_{\mathbf{PF}}, \mathbf{AF})$. Afterwards, the interventional constraint can be represented as $h(f, f_\delta)$. Our goal is to understand how minimal intervention in $\mathbf{PF}$ can lead to expected selection changes in $h(f, f_\delta)$. This can be formulated as an optimization problem, where we minimize the perturbations subject to a constraint on the change in the model's output as follows:

$$\min_{\delta_{\mathbf{PF}}} \quad \|\delta_{\mathbf{PF}}\|_2^2 + \lambda \|\delta_{\mathbf{PF}}\|_0$$
$$\text{s.t.} \quad \exists a \in \mathcal{A}, \forall a' \in \mathcal{A} - \{a\},$$
$$f(\mathbf{PF}, \mathbf{AF}(a')) - f(\mathbf{PF}, \mathbf{AF}(a)) < \varepsilon,$$
$$\exists a' \in \mathcal{A} - \{a\},$$
$$f_\delta(\mathbf{PF} + \delta_{\mathbf{PF}}, \mathbf{AF}(a')) - f_\delta(\mathbf{PF} + \delta_{\mathbf{PF}}, \mathbf{AF}(a)) > \varepsilon.$$
$$(14)$$

Here, we minimize $\delta_{\mathbf{PF}}$ from two perspectives: the overall magnitude of the intervention and the number of manipulated variables, inspired by the principle of Occam's Razor. $\|\delta_{\mathbf{PF}}\|_2$ represents the $\ell_2$-norm of the perturbations, indicating its magnitudes. The terms $\|\delta_{\mathbf{PF}}\|_0$ represents the $\ell_0$-norm, indicating the number of intervened problem features. The parameter $\lambda$ balances the trade-off between the magnitude and sparsity of the perturbations. The constraint $h(f, f_\delta, \varepsilon)$ in Eq. (14) states that there exists a candidate algorithm $a'$, such that after intervening on the problem

features, the optimal algorithm chosen by DAG-AS will change from algorithm $a$ to algorithm $a'$. The counterfactual explanation we derive is:

> If the features of problem $p$ are slightly decreased / increased by $|\delta_{\mathbf{PF}}|$, then the most suitable algorithm will change from algorithm $a$ to algorithm $a'$.

The $f_\delta$ in Eq. (14) is calculated based on a counterfactual computation procedure for DAG-AS model. Due to the nonlinear nature of the structural equations in DAG-AS, implemented through neural networks, we need to adjust the traditional steps accordingly, following three main steps: Abduction, Intervention, and Prediction (Pearl, 2010).

Given the observed evidence, i.e., the observed values of $\mathbf{PF}$ and $\mathbf{AF}$, we first infer the values of the exogenous noise terms $\epsilon_i^{\mathbf{PF}}$ and $\epsilon_i^{\mathbf{AF}}$ in the Abduction step. This step involves solving the neural network equations to find the noise terms that match the observed values, where the primary challenge is handling the nonlinear functions here. For complex multilayer neural networks, deriving an analytical solution is generally quite challenging. This is due to the fact that the nonlinear activation functions and the multilayer structure render the direct computation of the noise term a nonlinear optimization problem. As a consequence, we employ numerical methods, such as gradient descent, to iteratively solve for the solution here. The specific process can be found in **Appendix C**. After solving for all the exogenous variables corresponding to a given problem, we will modify the DAG-AS model based on the intervention values $\delta_{\mathbf{PF}}$. For all the variables that are to be intervened, we will delete the incoming edges from their parent nodes in the causal graph, in accordance with the do-operator, and then directly assign values to these features. The modified model is denoted as $f_\delta$. The intervention step alters the known causal graph and the underlying structural causal model, allowing us to make hypotheses about the model's behavior under manipulation. Finally, we can then solve for the counterfactual values of each variable in topological order. The results obtained at this stage represent the desired counterfactual outcomes.

### 3.3. Discussion on Explainability of DAG-AS

By applying counterfactual interventions and solving the optimization problems, we can gain a comprehensive understanding of the impact of problem features and the adaptation of algorithm features on the algorithm selection process in the DAG-AS model. This approach provides interpretable and actionable insights into the decision-making process, enabling us to identify key factors that influence the matching between problem features and algorithm features. It is important to note that although this paper does not investigate interventions on algorithm features, this does not

imply that such interventions are without significance. Let **a** denotes the values of **AF** before intervention. By examining $\mathbb{P}(S = 1 \mid \mathbf{PF}, do(\mathbf{AF} = \mathbf{a} + \delta_{\mathbf{AF}}))$, we can analyze how to adjust the optimal algorithms in the candidate set according to $\delta_{\mathbf{AF}}$ given the characteristics **PF** of the problem instance, ultimately leading to better solutions for the problem instance. Interventions on algorithm features hold practical significance for optimizing the algorithm beyond the candidate set.

On the other hand, we acknowledge that the explainability of our approach may not always be immediately intuitive. For instance, the algorithm features employed in ASlib are extracted from Abstract Syntax Tree, which are highly abstract and lack direct physical meaning. The utility of causal graph explanations and algorithm feature interventions hinges on the semantic interpretability of the features themselves. Specifically, when algorithm features correspond to intuitive, human-comprehensible attributes, such as the architectural components (e.g., number of layers, neuron counts) or hyperparameters (e.g., learning rate) of deep-learning models, the causal insights derived from DAG-AS become far more actionable. For example, if the causal graph identifies that certain problem features (e.g., input dimensionality) directly influence the number of neurons in a specific layer of a neural network, this insight can guide users in tailoring model architectures to problem characteristics. Such interpretability not only enhances human understanding of the algorithm selection process but also provides empirical guidance for optimizing candidate algorithms or debugging the DAG-AS model itself. This highlights the importance of feature semantics in unlocking the full potential of our causal framework for both explanation and algorithm design.

## 4. Experiments

**Benchmarks and Comparing Algorithms** (Detailed in **Appendix D.1**): This study employs the ASlib (Algorithm Selection Library) benchmark (Bischl et al., 2016) to evaluate various algorithm selection methods, providing a unified dataset with problem instances from multiple domains and their corresponding algorithm performance data. To validate DAG-AS's effectiveness, we test across all datasets that contain algorithm features, utilizing ten ASlib datasets detailed in Table 2 in Appendix D.1. Notably, algorithm features for the BNSL-2016 dataset were extracted using a large language model-based method (Wu et al., 2024). We measure algorithm performance using the PAR10 score, which compares actual running times to a predetermined cutoff time, assigning penalties for timeouts. A lower PAR10 score indicates a more effective algorithm selection method. Additionally, we evaluate five established methods, including ISAC (Kadioglu et al., 2010), MCC (multi-class classifica-

tion) (Xu et al., 2011), SATzilla11 (Xu et al., 2011), SNNAP (Collautti et al., 2013), and SUNNY (Amadini et al., 2014), alongside two baselines: the virtual best solver (VBS) and the single best solver (SBS), aiming to clarify their performance. Ideally, the performance of these methods should fall between those of SBS and VBS, with closer alignment to VBS indicating a more effective approach.

**Performance Comparison** (Detailed in **Appendix D.2**): In this experiment, each scenario was repeated 10 times for fair comparison, using $80\%$ of instances for training and $20\%$ for testing, with batch sizes of 1000 and 100, respectively. The results, as shown in Table 1, revealed that DAG-AS achieved the lowest PAR10 values in 8 out of 10 datasets, outperforming baseline algorithms and demonstrating robustness across various tasks, particularly in 5 datasets like GRAPHS-2015 and MAXSAT19-UCMS. This underscores the practical value of incorporating causality into algorithm selection. However, in datasets GLUHACK-18 and SAT03-16-INDU, DAG-AS lagged behind methods like MCC and SATzilla11, attributed to the difficulty of capturing causal relationships with limited training samples, leading to better performance by traditional correlation-based methods.

**Ablation Study** (Detailed in **Appendix D.3**): In this ablation study, we evaluated three variants of DAG-AS to assess the impact of the causal learning module and DAG design on model performance. The variants included: (1) Without Causality, where the causal module is removed, leading to direct matching of problem and algorithm features; (2) With Directed Cyclic Graph, retaining the causal module but allowing cycles in the graph; and (3) With Directed Acyclic Graph, the full version of DAG-AS with a constrained DAG structure. Using 10 datasets, results (shown in Figure 5 in Appendix D.3) indicate that the full DAG-AS variant consistently outperforms the others, especially on datasets like SAT11-INDU and SAT11-RAND, highlighting the DAG's critical role in capturing causal relationships and avoiding information loops. Conversely, the model without causality generally underperformed, particularly in datasets such as GRAPHS-2015 and MAXSAT19-UCMS, confirming that the absence of causal inference limits understanding of complex tasks. An exception was observed in SAT03-16-INDU, where focusing on correlations yielded better results due to challenges in capturing causal structures.

**Robustness against Distribution Shift** (Detailed in **Appendix D.4**): In this experiment, we evaluated the robustness of DAG-AS against various distribution shifts and compared its performance degradation to the best benchmark methods. We implemented three types of shifts: (1) Shift on Problem Distribution, where certain problem features were assigned higher sampling weights during training; (2) Shift on Optimal Algorithm Distribution, which involved prioritizing samples with optimal algorithms; and (3) Distribution

*Table 1.* Evaluation results on ASlib benchmarks with algorithm features.

| Scenario | VBS | SBS | ISAC | MCC | SATzilla11 | SNNAP | SUNNY | DAG-AS |
|---|---|---|---|---|---|---|---|---|
| BNSL-2016 | 211.09 | 8849.64 | $(6.32\pm1.42)\times10^3$ | $(3.79\pm1.38)\times10^3$ | $(2.00\pm0.76)\times10^3$ | $(37.37\pm3.46)\times10^3$ | $(3.86\pm1.29)\times10^3$ | $(\mathbf{1.81\pm0.57})\times10^3$ |
| GLUHACK-18 | 906.39 | 17829.03 | $(15.56\pm4.40)\times10^3$ | $(\mathbf{8.10\pm4.19})\times10^3$ | $(8.11\pm3.91)\times10^3$ | $(20.44\pm4.58)\times10^3$ | $(10.51\pm3.92)\times10^3$ | $(15.53\pm3.74)\times10^3$ |
| GRAPHS-2015 | $2.37\times10^6$ | $3.59\times10^7$ | $(8.20\pm3.20)\times10^6$ | $(11.23\pm3.79)\times10^6$ | $(8.90\pm3.18)\times10^6$ | $(64.32\pm46.91)\times10^6$ | $(7.56\pm3.50)\times10^6$ | $(\mathbf{5.42\pm2.13})\times10^6$ |
| MAXSAT19-UCMS | 126.93 | 8991.26 | $(4.20\pm2.01)\times10^3$ | $(4.00\pm1.40)\times10^3$ | $(3.77\pm1.35)\times10^3$ | $(9.43\pm1.44)\times10^3$ | $(3.37\pm1.91)\times10^3$ | $(\mathbf{2.15\pm1.04})\times10^3$ |
| SAT03-16-INDU | 823.02 | 5198.46 | $(4.73\pm0.84)\times10^3$ | $(4.47\pm0.63)\times10^3$ | $(\mathbf{4.27\pm0.58})\times10^3$ | $(10.41\pm1.88)\times10^3$ | $(4.46\pm0.78)\times10^3$ | $(8.15\pm0.73)\times10^3$ |
| SAT11-HAND | 1789.23 | 13028.87 | $(12.13\pm7.33)\times10^3$ | $(9.50\pm3.28)\times10^3$ | $(7.54\pm2.39)\times10^3$ | $(29.58\pm6.08)\times10^3$ | $(11.40\pm3.84)\times10^3$ | $(\mathbf{4.69\pm2.24})\times10^3$ |
| SAT11-INDU | 1108.52 | 9704.52 | $(8.03\pm3.06)\times10^3$ | $(6.40\pm3.51)\times10^3$ | $(6.79\pm3.05)\times10^3$ | $(15.21\pm3.84)\times10^3$ | $(6.54\pm3.61)\times10^3$ | $(\mathbf{3.06\pm1.36})\times10^3$ |
| SAT11-RAND | 1248.49 | 9433.79 | $(9.02\pm3.13)\times10^3$ | $(2.82\pm1.68)\times10^3$ | $(5.22\pm1.85)\times10^3$ | $(35.07\pm3.66)\times10^3$ | $(2.21\pm1.14)\times10^3$ | $(\mathbf{1.89\pm1.06})\times10^3$ |
| SAT18-EXP | 1705.43 | 11945.73 | $(10.39\pm4.41)\times10^3$ | $(9.76\pm4.17)\times10^3$ | $(8.95\pm3.16)\times10^3$ | $(44.09\pm3.06)\times10^3$ | $(8.11\pm3.65)\times10^3$ | $(\mathbf{8.07\pm1.49})\times10^3$ |
| TSP-LION2015 | 44.63 | 189.65 | $(1.12\pm0.35)\times10^3$ | $(2.67\pm0.48)\times10^3$ | $(1.84\pm0.42)\times10^3$ | $(10.22\pm1.61)\times10^3$ | $(0.78\pm0.16)\times10^3$ | $(\mathbf{0.48\pm0.21})\times10^3$ |

Shift by Removing Algorithms, where some candidate algorithms were omitted from the training data. Using nine ASlib benchmarks with uniformly provided algorithm features, we analyzed performance loss for each method under these shifts, as illustrated in Figure 6. DAG-AS consistently outperformed other algorithms across all shifts, particularly excelling in optimal algorithm distribution shifts, demonstrating its ability to leverage algorithm feature information effectively. While it maintained a competitive edge in most datasets during problem distribution shifts, it showed slight underperformance on the SAT18-EXP dataset. Nevertheless, DAG-AS remained robust, especially when algorithms were removed during training, as its causal learning module helped mitigate the impact of missing candidates.

**Demonstration of Model-level Explainability** (Detailed in **Appendix D.5**): In the algorithm selection scenario, deriving a causal graph like Figure 8 that reflects the relationships between problem and algorithm features enhances DAG-AS's accuracy and allows for meaningful causal interpretations. This study first analyzes feature importance in the causal graph by measuring the betweenness centrality (Barrat et al., 2004) of features (as shown in Figure 7), revealing that algorithm features generally have higher centrality values, indicating their significance in algorithm selection. Next, we explore causal relationships among algorithm features (as shown in Figure 8), finding interdependencies that vary across datasets, which helps identify predictive features and improve performance. Additionally, we assess the influence of problem features on algorithm features (as shown in Figure 9), noting that datasets with denser causal relationships, like MAXSAT19-UCMS and SAT11-HAND, correlate with better DAG-AS performance. In contrast, sparser relationships in datasets like SAT03-16-INDU can lead to underperformance. However, in simpler datasets with ample training data, such as TSP-LION2015, DAG-AS can still excel despite sparse causal relationships, highlighting its ability to focus on critical influences.

**Demonstration of Instance-level Explainability** (Detailed in **Appendix D.6**): DAG-AS constructs a causal graph that facilitates counterfactual explanations through targeted interventions. A demonstration experiment on the GRAPHS-2015 dataset illustrates this explainability by solving an opti-

mization problem to determine minimal intervention values that can alter selection results, under constraints of a maximum 20% feature intervention and a magnitude below 10%. The experiment identified 79 instances where the algorithm selection changed, with heatmaps Figure 10 showing total intervention magnitudes and specific feature interventions. For example, in Instance 1846, interventions on certain features shifted the selection from one candidate algorithm to another, highlighting the sensitivity of specific features to perturbations. Overall, only 1.36% of samples shifted selections, indicating DAG-AS's robust decision-making process. This analysis emphasizes DAG-AS's ability to provide interpretable insights and enhance transparency through causal graph-based reasoning.

## 5. Conclusion

In this study, we introduce a causal framework for algorithm selection, DAG-AS, which models the underlying mechanisms that determine algorithm suitability, addressing the limitations of existing methods. By focusing on the conditional relationships between problem and algorithm features, our approach enhances robustness against distribution changes and explainability of selection decisions. We propose a counterfactual explanation method that identifies minimal interventions necessary to achieve changes in algorithm selection decisions. Experimental results on the ASlib benchmark demonstrate that our model outperforms traditional techniques in both robustness and explainability. Future work could further enhance the efficiency of counterfactual interpretability methods. Additionally, by manipulating algorithm features, the explanation methods proposed in this paper may help uncover unknown algorithms, thereby facilitating the design of improved algorithms.

## Acknowledgements

This work was supported by the National Natural Science Foundation of China (Grant No. U21A20512 and 62306259), Research Grants Council of the Hong Kong SAR (Grant No. C5052-23G, PolyU25216423, PolyU15217424, PolyU15218622, PolyU15215623, and PolyU15229824), The Hong Kong Polytechnic University

(Grant No. P0043563, P0046094, P0053699, P0052694, and P0053758), and Natural Science Foundation of Chongqing (Grant No. CSTB2022NSCQ-MSX1285).

## Impact Statement

This paper presents work whose goal is to advance the field of Machine Learning. There are many potential societal consequences of our work, none which we feel must be specifically highlighted here.

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

# A. Background

## A.1. Background of Algorithm Selection

With the rapid advancement and widespread adoption of information technology, researchers in various fields have designed a range of algorithms tailored to specific types of problems, each showing different levels of performance across various instances of those problems. Due to the diversity of problems, a single algorithm often fails to perform well across all scenarios (Pihera & Musliu, 2014; Xu et al., 2011). However, selecting the best algorithm for a specific problem instance is a complex and time-consuming process. To meet this challenge, research in algorithm selection (Rice, 1976) has become increasingly important. The primary goal is to identify the most suitable algorithm from a pool of candidates, ultimately enhancing performance on specific tasks, whether in terms of accuracy (Ruhkopf et al., 2023) and time efficiency (Kostovska et al., 2023). In recent years, algorithm selection methods have become essential for tackling complex problems and driving technological advancements, with applications across various fields, such as machine learning (Luo, 2016) and combinatorial optimization (Pihera & Musliu, 2014).

Early algorithm selection methods sought suitable meta-features to characterize a given task, such as the statistical characteristics of the input dataset (Hutter et al., 2014) or probing features derived from the short execution of well-known solver (Nudelman et al., 2004). By exploring the relationship between these meta-features and the performance of different algorithms, machine learning techniques were employed to meta-learn which algorithm performs best on new problem instances. These methods often formulate the algorithm selection task as either a performance regression or a multi-class classification problem. Specifically, regression-based techniques (Xu et al., 2008) directly predict the performance of each algorithm based on the problem feature set. Conversely, multi-class classification approaches (Cunha et al., 2018; Abdulrahman et al., 2018) assign scores to algorithms based on their relative suitability for a specific problem and select the most appropriate algorithm based on these scores. In addition, several other methodologies have been proposed. For instance, some studies formalize algorithm selection as a collaborative filtering problem (Mısır & Sebag, 2017; Fusi et al., 2018) and utilize a sparse matrix with performance data available for only a few algorithms on each problem instance. And similarity-based methods (Amadini et al., 2014; Kadioglu et al., 2010) select algorithms based on the similarity between the current problem instance and previously encountered instances. There are also some hybrid methods (Hanselle, 2020; Fehring et al., 2022) combine multiple techniques above.

On the other hand, some studies (Hough & Williams, 2006; Tornede et al., 2020; Hilario et al., 2009; Pulatov et al., 2022; de Nobel et al., 2021; Wu et al., 2024) have recognized the role of algorithm meta-features, such as hyperparameters (Hough & Williams, 2006; Tornede et al., 2020), model structure information (Hilario et al., 2009), code-related statistical features (Pulatov et al., 2022), abstract syntax tree features (Pulatov et al., 2022), and and large language model-extracted code features (Wu et al., 2024). Empirical studies (Pulatov et al., 2022; Wu et al., 2024) indicate that utilizing algorithm meta-features can enhance a model's generalization across different algorithms and, in certain cases, improve the accuracy of algorithm selection. Models based on algorithm features typically concatenate problem features and algorithm features and employ machine learning algorithms to either regress algorithm performance, predict the most suitable algorithm, or calculate the matching degree between problem features and algorithm features to make compatibility decisions.

In recent years, research on algorithm selection has focused on various domains, including black-box optimization problems (Kostovska et al., 2022; Huerta et al., 2022), natural science issues (Chen et al., 2023), software engineering (Richter et al., 2020), and machine learning (Arena et al., 2024; Dagan et al., 2024). Some studies have also explored feature-free approaches (Alissa et al., 2023), which are mainly applied to specific senoritas, such as traveling salesman problem (Zhao et al., 2021) and continuous optimization problem (Seiler et al., 2024).

## A.2. Background of Causal Learning

In this subsection, we introduce the fundamental concepts of causal learning, which are essential for understanding the algorithm selection method based on causality. Causal relationships describe the cause-effect connections between two events, reflecting the internal mechanisms and rules within a system. A Causal Bayesian Network (CBN) is a framework that uses a directed acyclic graph (DAG) to explain causal relationships. Its definition is as follows:

**Definition A.1. Causal Bayesian Network** (CBN (Pearl, 2009)). A Causal Bayesian Network is a directed acyclic graph $\mathcal{G} = \{\mathcal{V}, \mathcal{E}\}$ where nodes represent random variables in the set $\mathcal{V}$, and edges in the set $\mathcal{E}$ represent causal relationships between these variables. Each node $V_i \in \mathcal{V}$ is associated with a conditional probability distribution given its parents $\mathbf{Pa}(V_i)$

in the graph. The joint probability distribution $\mathbb{P}$ over $\mathcal{V}$ can be factorized as:

$$\mathbb{P}(\mathcal{V}) = \prod_{V_i \in \mathcal{V}} \mathbb{P}(V_i \mid \mathbf{Pa}(V_i)) \tag{15}$$

A Random Control Trial is an experimental method for causal inference, regarded as the gold standard for inferring causality, but it is not applicable to the observational data in our scenario. To infer causality from observational data, researchers have developed data-driven causal inference models called Structural Causal Models (SCMs). SCMs represent causal relationships through causal graphs (DAGs) and structural equations. The causal graph is a graphical qualitative representation of causal relationships, while the structural equations quantitatively represent these relationships.

**Definition A.2. Structural Causal Model** (SCM (Pearl, 2012)). A Structural Causal Model (SCM) consists of a set of structural equations and a causal graph $\mathcal{G}$ that represents the dependencies between variables. Each structural equation specifies how a particular variable is generated by its parent variables and an error term that captures all unobserved influences. Formally, an SCM for can be expressed as:

$$V_i = f_i(\mathbf{Pa}(V_i), U_i), \quad \forall V_i \in \mathcal{V}, \tag{16}$$

where $V_i$ is a variable, $\mathbf{Pa}(V_i)$ are the parents of $V_i$ in the causal graph, and $U_i \in \mathcal{U}$ is an exogenous error term.

SCMs can represent causal relationships among complex multidimensional variables. The variables in $\mathcal{V}$ and $\mathcal{U}$ are called endogenous and exogenous variables, respectively. Exogenous variables belong to the external environment of the model and appear as noise terms, representing influences on endogenous variables that are not influenced by them. Each endogenous variable is a descendant of at least one exogenous variable, so the root nodes in the causal graph over $\mathcal{V}$ in SCM have only one parent variable, which is an exogenous variable.

The graphical model and structural equations enable SCMs to conveniently compute causal effects and counterfactual queries from observational data. Pearl developed a comprehensive theory and methods by combining SCMs with do-calculus:

**Definition A.3. Causal Effect and Intervention.** (Pearl, 2009) The intervention do(T=t) denotes setting the value of variable $T$ to $t$, denoted as do(t). The causal effect of variable $T$ on variable $Y$ is a probability distribution function from $T$ to $Y$, represented using the do operator as $P(Y = y \mid do(T = t))$, abbreviated as $P(y \mid do(t))$.

In SCMs, interventions are represented by modifying the structural equations. The intervention $do(T = t)$ replaces the equation $T = f_T(\mathbf{Pa}(T), U_t)$ with $T = t$ in the SCM, and $T = t$ is substituted in the remaining equations. This demonstrates the modular nature of causal models, where each function equation represents an independent causal mechanism. The intervention $do(T = t)$ changes only the assignment function containing $T$, leaving the assignment functions of other variables unchanged.

**Definition A.4. Counterfactuals.** (Pearl, 2009) Let $T$ and $Y$ be two variables in $\mathcal{V}$. The counterfactual statement "if $T = t$, then $Y$ would be $y$ given $U = u$" is represented as $Y_{T=t}(u) = y$, where $Y_{T=t}(u)$ is the potential outcome of $Y$ given $T = t$ under the specific context $U = u$.

In any SCM, if the counterfactual assumption is $T = t$, the modified model is denoted as SCM$_t$. The counterfactual value $Y_t(u)$ in the model SCM is defined as the value of $Y$ obtained in the new model SCM$_t$ when $T = t$:

$$Y_t(u) = Y_{SCM_t}(u) \tag{17}$$

This definition holds even if $T$ and $Y$ are sets of variables rather than individual variables. In the modified model SCM$_t$, the values of the variables should satisfy the new model's specifications.

Understanding these fundamental concepts is crucial for developing and implementing algorithms that leverage causal knowledge. The formal definitions and mathematical frameworks provide the basis for more advanced applications in causal learning-based algorithm selection.

## B. Proof of Theorem 2.4

**Theorem B.1.** *Assume that the joint density function of $\mathbb{P}(\mathbf{PF}, \mathbf{AF})$ is continuous and strictly positive on a compact and convex subset of $\mathbb{R}^{\mathbf{PF} \cup \mathbf{AF}}$, and zero elsewhere. And there exists a DAG $\mathcal{G}$ such that $\mathbb{P}(\mathbf{PF}, \mathbf{AF} \mid S = 1)$ can be factorized as Eq.*

*(15) according to $\mathcal{G}$. Under Assumption 2.1, any variable $\boldsymbol{AF}_i \in \boldsymbol{AF}$ is not an exogenous variable in $\mathcal{G}$, and there exists a continuous function set $f = \{f_1, f_2, \cdots, f_{|\boldsymbol{AF}|}\}$ with compact support in $\mathbb{R}^{|\boldsymbol{PF}| \times [0,1]}$ such that each $\mathbb{P}(\boldsymbol{AF}_i|\boldsymbol{Pa}(\boldsymbol{AF}_i), S = 1)$ equals the corresponding $f_i$ with uniform distributed noise on $[0, 1]$.*

*Proof.* Since the in-degree of any $\mathbf{AF}_i$ satisfies $\deg^-(\mathbf{AF}_i) > 0$, i.e., $|\mathbf{Pa}(\mathbf{AF}_i)| \geq 1$, we can easily prove that there exists $\mathbf{AF}_i \in \mathbf{AF}$ such that its parent set is completely composed of problem features, i.e., $\mathbf{Pa}(\mathbf{AF}_i) \subseteq \mathbf{PF}$. Assume that for all $\mathbf{AF}_i \in \mathbf{AF}$, $\mathbf{Pa}(\mathbf{AF}_i) \not\subseteq \mathbf{PF}$. This means that for every $\mathbf{AF}_i$, there exists at least one algorithm feature $X_j$ such that $X_j \in \mathbf{Pa}(\mathbf{AF}i)$ and $X_j \notin \mathbf{PF}$. We can then construct a causal chain $X_N \to \cdots \to X_2 \to X_1$, where each $X_i$ is an algorithm feature and $X_i \in \mathbf{Pa}(X_{i-1})$. However, due to the acyclicity requirement, no variable can be its own ancestor or descendant. Therefore, when $N > |\mathbf{AF}|$, there will be no new algorithm features to satisfy $\deg^-(X_N) > 0$, which contradicts the assumption that the in-degree of any $\mathbf{AF}_i$ is positive. Therefore, we can conclude that $\exists \mathbf{AF}_i$ s.t. $\mathbf{Pa}(\mathbf{AF}_i) \subseteq \mathbf{PF}$.

Let us divide the nodes in $\mathcal{G}$ into two sets, $\mathbf{A}$ and $\mathbf{B}$, where $\mathbf{A}$ is initialized as $\mathbf{PF}$ and $\mathbf{B}$ is initialized as $\mathbf{AF}$. Next, we find the first node $\mathbf{AF}_i$ in $\mathbf{B}$ such that $\mathbf{Pa}(\mathbf{AF}_i) \subseteq \mathbf{A}$, and perform the following operations:

$$\mathbf{A} \leftarrow \mathbf{A} \cup \mathbf{AF}_i, \ \mathbf{B} \leftarrow \mathbf{B} - \mathbf{AF}_i \tag{18}$$

We then repeat this process to find the next $\mathbf{AF}_i$ in $\mathbf{B}$ such that $\mathbf{Pa}(\mathbf{AF}_i) \subseteq \mathbf{A}$, and so on. This gives us a topological ordering $\pi : \mathbf{AF} \to \{1, 2, \dots, |\mathbf{AF}|\}$. Along this topological ordering $(\mathbf{AF}_{\pi(1)}, \mathbf{AF}_{\pi(2)}, \dots, \mathbf{AF}_{\pi(|\mathbf{AF}|)})$, we will now use mathematical induction to prove as follows:

For $\mathbf{AF}_{\pi(1)}$, we can construct a function to describe the corresponding decomposed distribution: $\mathbb{P}(\mathbf{AF}_{\pi(1)}|\mathbf{Pa}(\mathbf{AF}_{\pi(1)}), S = 1)$. Consider the conditional cumulative distribution function:

$$F_1(\mathbf{AF}_{\pi(1)}|\mathbf{Pa}(\mathbf{AF}_{\pi(1)}) = z_1, , S = 1) = \mathbb{P}(\mathbf{AF}_{\pi(1)} < x_1|\mathbf{Pa}(\mathbf{AF}_{\pi(1)}) = z_1, , S = 1). \tag{19}$$

where $x_i$ and $z_i$ are arguments of the function $F_i$, denoting the value of $\mathbf{AF}_{\pi(i)}$ and $\mathbf{Pa}(\mathbf{AF}_{\pi(i)})$. Since the joint density function is strictly positive, $F_1$ is strictly continues and monotonic wrt $x_1$. Therefore, its inverse, the quantile function $F_1^{-1}(\mathbf{Pa}(\mathbf{AF}_{\pi(1)}), \epsilon) : \text{dom}(\mathbf{Pa}(\mathbf{AF}_{\pi(1)})) \times [0, 1]$, where $\epsilon$ represents the noise term in the causal model and $\text{dom}(*)$ denotes the domain of the corresponding variable, satisfies the properties in Theorem 1.

Assume that for $\mathbf{AF}_{\pi(i-1)}$, we can find a function that satisfies Theorem 1. We now prove the case for $\mathbf{AF}_{\pi(i)}$: Take the conditional cumulative distribution function $F_i(\mathbf{AF}_{\pi(i)}|\mathbf{Pa}(\mathbf{AF}_{\pi(i)}) = z_i, S = 1)$ and its inverse $F_i^{-1}(\mathbf{Pa}(\mathbf{AF}_{\pi(i)}), \epsilon) : \text{dom}(\mathbf{Pa}(\mathbf{AF}_{\pi(i)})) \times [0, 1]$. By the topological ordering rule, all of the parent nodes of $\mathbf{AF}_{\pi(i)}$ are included in $\mathbf{A}$. Therefore, $F_i^{-1}(\mathbf{Pa}(\mathbf{AF}_{\pi(i)}), \epsilon) = F_i^{-1}(\{F_j^{-1}\}_{j=1}^{i-1}, \epsilon)$. Since $F_j^{-1}$ ($j \in \{1, \dots, i-1\}$) can be found to satisfy Theorem 1, $F_i^{-1}(\{F_j^{-1}\}_{j=1}^{i-1}, \epsilon)$ also satisfies the properties in Theorem 1. (Q.E.D.) $\square$

## C. Calculate the Exogenous Variables for Counterfactual Explainability

Given the observed evidence, i.e., the observed values of $\mathbf{PF}$ and $\mathbf{AF}$, we first infer the values of the exogenous noise terms $\epsilon_i^{\mathbf{PF}}$ and $\epsilon_i^{\mathbf{AF}}$ in the Abduction step. This step involves solving the neural network equations to find the noise terms that match the observed values, where the primary challenge is handling the nonlinear functions here. For complex multilayer neural networks, deriving an analytical solution is generally quite challenging. This is due to the fact that the nonlinear activation functions and the multilayer structure render the direct computation of the noise term a nonlinear optimization problem. As a consequence, we employ numerical methods, such as gradient descent, to iteratively solve for the solution here. For the noise term $\epsilon_i^{\mathbf{PF}}$ of the problem feature $\mathbf{PF}_i$, we need to solve for the gradient $\nabla \epsilon_i^{\mathbf{PF}}$ in order to iteratively compute the value of $\epsilon_i^{\mathbf{PF}}$ with rate $\eta$, i.e., the $(t + 1)$-th iteration is given by: $\epsilon_i^{\mathbf{PF}}(t + 1) = \epsilon_i^{\mathbf{PF}}(t) - \eta \nabla \epsilon_i^{\mathbf{PF}}$. Since the loss function for the prediction of $\mathbf{PF}_i$ can be represented as:

$$\mathcal{L}_{\mathbf{PF}} = \frac{1}{2}\|\mathbf{PF}_i - \hat{\mathbf{PF}}_i\|^2, \tag{20}$$

the gradient $\nabla \epsilon_i^{\mathbf{PF}}$ can be expressed as:

$$\nabla \epsilon_i^{\mathbf{PF}} = \frac{\partial \mathcal{L}_{\mathbf{PF}}}{\partial \epsilon_i^{\mathbf{PF}}} = \left( \frac{\partial \mathcal{L}_{\mathbf{PF}}}{\partial [\hat{\mathbf{PF}}; \epsilon_i^{\mathbf{PF}}]} \right)_{|\mathbf{PF}|+1}, \tag{21}$$

where $|\mathbf{PF}| + 1$ denotes the focused dimension of $\partial \mathcal{L}_{\mathbf{PF}}/\partial[\hat{\mathbf{PF}}; \epsilon_i^{\mathbf{PF}}]$. Let $\mathbf{z}^{(l)}$ denotes the input to the activation function at the $l$-th layer, then

$$\frac{\partial \mathcal{L}_{\mathbf{PF}}}{\partial[\hat{\mathbf{PF}}; \epsilon_i^{\mathbf{PF}}]} = W_1^{(i)T} \cdot \frac{\partial \mathcal{L}_{\mathbf{PF}}}{\partial \mathbf{z}^{(1)}}, \tag{22}$$

where the value of $\frac{\partial \mathcal{L}_{\mathbf{PF}}}{\partial \mathbf{z}^{(1)}}$ could be obtained by the following recurrence relation:

$$\begin{aligned}
\frac{\partial \mathcal{L}_{\mathbf{PF}}}{\partial \mathbf{z}^{(L)}} &= (\mathbf{PF}_i - \hat{\mathbf{PF}}_i)\sigma^{(L)'}(\mathbf{z}^{(L)}) \\
\frac{\partial \mathcal{L}_{\mathbf{PF}}}{\partial \mathbf{z}^{(l)}} &= \left(\mathbf{W}^{(l+1)T} \frac{\partial \mathcal{L}_{\mathbf{PF}}}{\partial \mathbf{z}^{(l+1)}}\right) \odot \sigma^{(l)'}(\mathbf{z}^{(l)})
\end{aligned} \tag{23}$$

For the noise term $\epsilon_i^{\mathbf{AF}}$ of the algorithm feature $\mathbf{AF}_i$, we need to replace the $\mathcal{L}_{\mathbf{PF}}$ with $\mathcal{L}_{\mathbf{AF}} = \frac{1}{2}\|\mathbf{AF}_i - \hat{\mathbf{AF}}_i\|^2$,, and the gradient $\nabla \epsilon_i^{\mathbf{AF}}$ can be expressed as:

$$\nabla \epsilon_i^{\mathbf{AF}} = \frac{\partial \mathcal{L}_{\mathbf{AF}}}{\partial \epsilon_i^{\mathbf{AF}}} = \left(\frac{\partial \mathcal{L}_{\mathbf{AF}}}{\partial[\hat{\mathbf{PF}}; \hat{\mathbf{AF}}; \epsilon_i^{\mathbf{AF}}]}\right)_{|\mathbf{PF} \cup \mathbf{AF}|+1}, \tag{24}$$

In addition to the numerical methods mentioned above, we also provide the analytical solution for a single-layer linear network, which can be used for the rapid computation of the noise term in the context of linear modeling:

$$\begin{aligned}
\epsilon_i^{\mathbf{PF}} &= \frac{\mathbf{W}_d^T \mathbf{PF}_i - \mathbf{W}_d^T \mathbf{W}_{\neg d}\mathbf{PF}}{\mathbf{W}_d^T \mathbf{W}_d} \\
\epsilon_i^{\mathbf{AF}} &= \frac{\mathbf{W}_d^T \mathbf{AF}_i - \mathbf{W}_d^T \mathbf{W}_{\neg d}[\mathbf{PF}; \mathbf{AF}]}{\mathbf{W}_d^T \mathbf{W}_d}
\end{aligned} \tag{25}$$

where $\mathbf{W}_d$ denotes the $d$-th column of the unique parameter matrix $\mathbf{W}$ in the network, and $\mathbf{W}_{\neg d}$ represents the matrix $\mathbf{W}$ with the $d$-th dimension removed. After solving for all the exogenous variables corresponding to a given problem, we will modify the DAG-AS model based on the provided intervention information.

## D. Detailed Experiments

In this section, we evaluate the performance of our proposed DAG-AS algorithm across multiple benchmark datasets from the ASlib library. The experimental analysis is designed to comprehensively assess the effectiveness of DAG-AS in various aspects, including its overall performance, robustness, and explainability. We conduct five major types of experiments to ensure a thorough evaluation: We first perform a performance comparison to benchmark DAG-AS against five classic algorithm selection methods across ten datasets in Section D.2. Then, an ablation study is carried out in Section D.3 to examine the impact of causal structures on the performance of DAG-AS, comparing results with and without the inclusion of causal graphs (both directed and undirected). We investigate the generalization ability of DAG-AS in Section D.4 by evaluating its performance under different types of distribution shifts. Finally, a causal graph analysis is conducted to delve into the causal relationships identified by DAG-AS for each dataset in Section D.5, and the explainability of DAG-AS is showcased by demonstrating its ability to generate counterfactual explanations using the GRAPHS-2015 dataset in Section D.6. Through these experiments, we aim to demonstrate not only the superior performance of DAG-AS but also its robustness and explainability in the field of algorithm selection.

### D.1. ASlib Benchmarks and Comparing Algorithms

**Evaluation Benchmarks:** The ASlib (Algorithm Selection Library) benchmark (Bischl et al., 2016) is a standardized dataset designed to provide a unified benchmark for evaluating and comparing the performance of various algorithm selection methods. As one of the most widely used benchmarks in the field of algorithm selection, ASlib encompasses problem instances from diverse domains along with their corresponding algorithm performance data. Each problem instance is described by a set of features that reflect the properties and constraints of the problem, with some datasets also including algorithm features. Given that this paper aims to uncover the causal relationships between problem features and algorithm features, we validate the effectiveness of DAG-AS by testing across all available datasets with algorithm features. The ten

ASlib datasets cover a range of domains and scales, with statistical information presented in Table 2. Notably, the algorithm features for the BNSL-2016 dataset are not provided by ASlib; instead, they were extracted from the code of candidate algorithms using the large language model-based feature extraction method proposed in (Wu et al., 2024).

*Table 2.* The statistical property of experimental benchmarks.

| ASlib Scenario | $|\mathcal{P}|$ | $|\mathcal{A}|$ | $|\boldsymbol{PF}|$ | $|\boldsymbol{AF}|$ | Cutoff Time |
|---|---|---|---|---|---|
| BNSL-2016 | 1179 | 8 | 86 | 75 | 7200 |
| GLUHACK-18 | 353 | 8 | 50 | 75 | 5000 |
| GRAPHS-2015 | 5725 | 4 | 35 | 75 | 100000000 |
| MAXSAT19-UCMS | 572 | 7 | 54 | 75 | 1800 |
| SAT03-16-INDU | 2000 | 8 | 483 | 75 | 5000 |
| SAT11-HAND | 296 | 11 | 115 | 75 | 5000 |
| SAT11-INDU | 300 | 18 | 115 | 75 | 5000 |
| SAT11-RAND | 600 | 8 | 115 | 75 | 5000 |
| SAT18-EXP | 353 | 37 | 50 | 75 | 5000 |
| TSP-LION2015 | 31060 | 4 | 122 | 75 | 3600 |

The involved scenarios in this paper focus on the solution time of candidate algorithms, and therefore, we use the PAR10 score to measure the performance of different algorithms. Specifically, the PAR10 for instance $p$ is calculated as follows:

$$\mathrm{PAR}\,10(p) = \begin{cases} t_p & \text{if } t_p \leqslant C \\ 10 \cdot C & \text{else} \end{cases} . \tag{26}$$

For each problem instance $p$, the actual running time $t_p$ of the selected algorithm is compared to a predetermined cutoff time $C$, as provided in Table 2. If the selected algorithm finds a solution within the cutoff time, the actual running time is recorded; otherwise, a penalty of ten times the cutoff time, $10 \cdot C$, is incurred. Finally, the PAR10 score is obtained by averaging the results across all problem instances. The PAR10 score takes into account both the solution time and timeout situations, with a lower PAR10 score indicating a more effective algorithm selection method.

**Comparing Algorithms:** This study evaluates five established algorithm selection methods: ISAC (Kadioglu et al., 2010), MCC (multi-class classification) (Xu et al., 2011), SATzilla11 (Xu et al., 2011), SNNAP (Collautti et al., 2013), and SUNNY (Amadini et al., 2014). In addition to these methods, we also compare two key performance benchmarks: the virtual best solver (VBS) and the single best solver (SBS). VBS represents the optimal approach, as it selects the best algorithm for each specific problem instance. In contrast, SBS is a straightforward method that picks the algorithm with the best overall performance without differentiating between instances. By evaluating the algorithm selection methods against SBS and VBS, we can gain a clearer understanding of their performance. Ideally, the performance of these methods should fall between those of SBS and VBS, with closer alignment to VBS indicating a more effective approach.

### D.2. Performance Comparison

In this experiment, we employed multiple ASlib benchmark datasets to evaluate the performance of DAG-AS on algorithm selection tasks. To ensure a fair comparison between the proposed DAG-AS and existing classical methods, each algorithm selection scenario was repeated 10 times. The training set consisted of $80\%$ of the samples randomly selected from the dataset. The batch sizes for training and testing were set to 1000 and 100, respectively. Both problem features and algorithm features were fed into the model, and the final performance was compared based on the PAR10 scores of the different models, as shown in Table 3.

By analyzing the overall results, we found that DAG-AS achieved the best performance in 8 out of the 10 datasets, obtaining the lowest PAR10 values. This indicates that DAG-AS outperformed other baseline algorithms in most tasks, demonstrating strong robustness and wide applicability to different types of algorithm selection problems. In particular, DAG-AS significantly outperformed other methods in datasets such as GRAPHS-2015, MAXSAT19-UCMS, SAT11-HAND, SAT11-INDU, and TSP-LION2015. This strongly highlights the practical value of incorporating causality into algorithm selection. By leveraging a causal graph structure, DAG-AS enhances the model's understanding of complex problems and decision-making capabilities, effectively utilizing the causal relationships between problem features and algorithm features to improve the accuracy of algorithm selection.

*Table 3.* Evaluation results on ASlib benchmarks with algorithm features.

| Scenario | VBS | SBS | ISAC | MCC | SATzilla11 | SNNAP | SUNNY | DAG-AS |
|---|---|---|---|---|---|---|---|---|
| BNSL-2016 | 211.09 | 8849.64 | $(6.32\pm1.42)\times10^3$ | $(3.79\pm1.38)\times10^3$ | $(2.00\pm0.76)\times10^3$ | $(37.37\pm3.46)\times10^3$ | $(3.86\pm1.29)\times10^3$ | $(\mathbf{1.81\pm0.57})\times10^3$ |
| GLUHACK-18 | 906.39 | 17829.03 | $(15.56\pm4.40)\times10^3$ | $(\mathbf{8.10\pm4.19})\times10^3$ | $(8.11\pm3.91)\times10^3$ | $(20.44\pm4.58)\times10^3$ | $(10.51\pm3.92)\times10^3$ | $(15.53\pm3.74)\times10^3$ |
| GRAPHS-2015 | $2.37\times10^6$ | $3.59\times10^7$ | $(8.20\pm3.20)\times10^6$ | $(11.23\pm3.79)\times10^6$ | $(8.90\pm3.18)\times10^6$ | $(64.32\pm46.91)\times10^6$ | $(7.56\pm3.50)\times10^6$ | $(\mathbf{5.42\pm2.13})\times10^6$ |
| MAXSAT19-UCMS | 126.93 | 8991.26 | $(4.20\pm2.01)\times10^3$ | $(4.00\pm1.40)\times10^3$ | $(3.77\pm1.35)\times10^3$ | $(9.43\pm1.44)\times10^3$ | $(3.37\pm1.91)\times10^3$ | $(\mathbf{2.15\pm1.04})\times10^3$ |
| SAT03-16-INDU | 823.02 | 5198.46 | $(4.73\pm0.84)\times10^3$ | $(4.47\pm0.63)\times10^3$ | $(\mathbf{4.27\pm0.58})\times10^3$ | $(10.41\pm1.88)\times10^3$ | $(4.46\pm0.78)\times10^3$ | $(8.15\pm0.73)\times10^3$ |
| SAT11-HAND | 1789.23 | 13028.87 | $(12.13\pm7.33)\times10^3$ | $(9.50\pm3.28)\times10^3$ | $(7.54\pm2.39)\times10^3$ | $(29.58\pm6.08)\times10^3$ | $(11.40\pm3.84)\times10^3$ | $(\mathbf{4.69\pm2.24})\times10^3$ |
| SAT11-INDU | 1108.52 | 9704.52 | $(8.03\pm3.06)\times10^3$ | $(6.40\pm3.51)\times10^3$ | $(6.79\pm3.05)\times10^3$ | $(15.21\pm3.84)\times10^3$ | $(6.54\pm3.61)\times10^3$ | $(\mathbf{3.06\pm1.36})\times10^3$ |
| SAT11-RAND | 1248.49 | 9433.79 | $(9.02\pm3.13)\times10^3$ | $(2.82\pm1.68)\times10^3$ | $(5.22\pm1.85)\times10^3$ | $(35.07\pm3.66)\times10^3$ | $(2.21\pm1.14)\times10^3$ | $(\mathbf{1.89\pm1.06})\times10^3$ |
| SAT18-EXP | 1705.43 | 11945.73 | $(10.39\pm4.41)\times10^3$ | $(9.76\pm4.17)\times10^3$ | $(8.95\pm3.16)\times10^3$ | $(44.09\pm3.06)\times10^3$ | $(8.11\pm3.65)\times10^3$ | $(\mathbf{8.07\pm1.49})\times10^3$ |
| TSP-LION2015 | 44.63 | 189.65 | $(1.12\pm0.35)\times10^3$ | $(2.67\pm0.48)\times10^3$ | $(1.84\pm0.42)\times10^3$ | $(10.22\pm1.61)\times10^3$ | $(0.78\pm0.16)\times10^3$ | $(\mathbf{0.48\pm0.21})\times10^3$ |

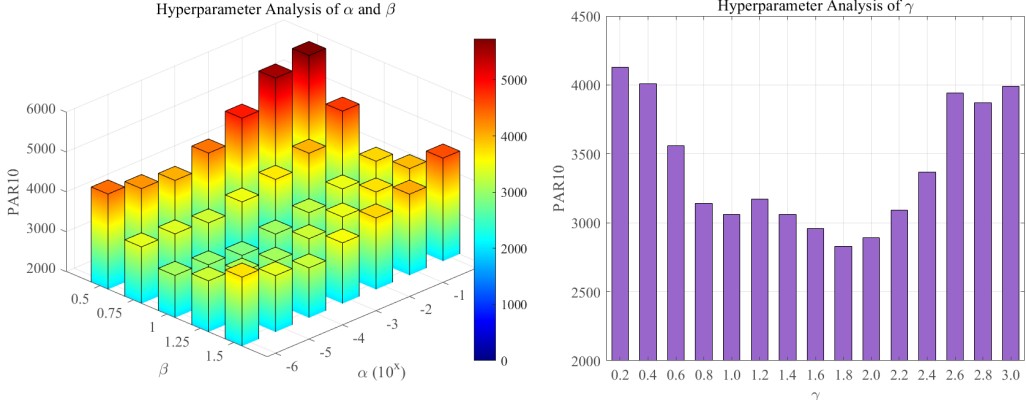

*Figure 4.* Hyperparameter analysis on SAT11-INDU.

At the same time, we also observed that in some datasets, such as GLUHACK-18 and SAT03-16-INDU, DAG-AS did not perform as well as other methods, with MCC and SATzilla11 achieving the best performance, respectively. The main reason is that causal relationships between problem features and algorithm features in these two datasets are more difficult to capture. With a limited number of training samples, DAG-AS was unable to sufficiently learn the patterns associated with these causal relationships. Instead, the correlations exhibited in the data were more easily captured, which led to DAG-AS underperforming compared to traditional correlation-based methods on these datasets.

**Hyperparameter analysis:** We conducted a hyperparameter analysis on the SAT11-INDU dataset to investigate the impact of the three hyperparameters in Eq.(15) on the performance of DAG-AS. In the first experiment, we analyzed the balance between the reconstruction loss and the two causal learning constraints by varying $\alpha$ and $\beta$, while fixing $\gamma = 1$. The second experiment focused on the balance between the causal learning loss and the algorithm selection loss by adjusting $\gamma$, with $\beta = 1$ and $\alpha = 0.0001$ kept constant. Results are shown in Figure 4. From the left-hand figure, the color-coded visualization (cooler colors indicating lower PAR10 scores) reveals a range of $\alpha$ and $\beta$ values where DAG-AS sustains good performance. For the right-hand figure, as $\gamma$ changes, the PAR10 score does not exhibit extreme fluctuations, indicating DAG-AS is not overly sensitive to $\gamma$ within the tested range. Notably, when $\gamma$ is around 1.4–2.2, the PAR10 score attains relatively low values, showcasing excellent performance. Overall, these results demonstrate that DAG-AS maintains relatively good performance across a broad spectrum of these hyperparameters, underscoring the model's robustness. This robustness is vital as it minimizes reliance on meticulous hyperparameter tuning, enhancing DAG-AS's adaptability to diverse problem scenarios.

### D.3. Ablation Study

In this ablation study, we compare three variants of DAG-AS to further validate the impact of the causal learning module and the DAG design on model performance. The three model variants compared in the experiment are as follows: (1) Without Causality: The causal learning module is removed, and algorithm selection is performed directly based on matching problem features and algorithm features. (2) With Directed Cyclic Graph: The causal learning module is retained, but cycles are

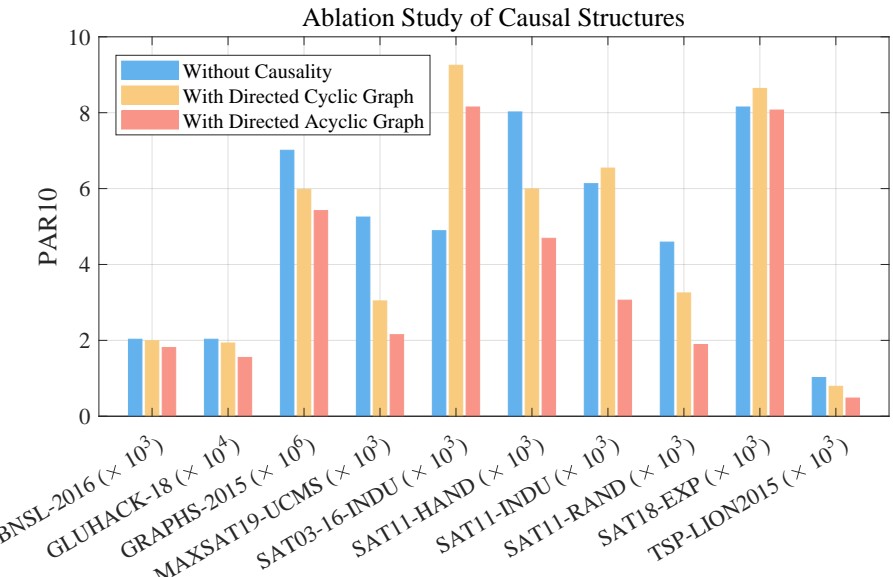

*Figure 5.* Ablation study on ASlib benchmarks.

allowed in the causal graph (i.e., the causal graph is a directed cyclic graph), meaning the learning process is not constrained by the DAG structure. (3) With Directed Acyclic Graph: This is the full version of DAG-AS, where the causal graph is constrained to be a DAG, ensuring no cycles exist between causal relations, enhancing the modeling of the relationships between problem and algorithm features through causal inference. The 10 datasets split into training and test sets, as well as the overall experimental process, follows the same procedure as in previous experiments.

As shown in Figure 5, we can clearly observe that the case with directed acyclic graph (i.e., the entire DAG-AS) demonstrates superior performance on most datasets, which indicates that the identification of causal relationships and the DAG structure design significantly enhance model performance in algorithm selection tasks. Compared to the case with directed cyclic graph, DAG-AS shows a notable advantage across all datasets, especially on the SAT11-INDU and SAT11-RAND datasets, suggesting that the DAG is essential for capturing causal relationships in complex tasks. The acyclic nature of the causal structure ensures the unidirectionality of causal relationships, which is crucial for avoiding information loops and redundancy, helping the DAG-AS learn more accurate relational patterns. On the other hand, in the case without causality, where the causal learning module is removed, the model generally performs worse across most datasets. The performance gap is particularly evident on GRAPHS-2015, MAXSAT19-UCMS, SAT11-HAND, SAT11-INDU, and SAT11-RAND datasets, indicating that the absence of causal inference greatly limits the model's understanding of the complexity of the task. However, there is one exception: on the SAT03-16-INDU dataset, the model without the causal learning module outperforms DAG-AS, which further validates that the performance deficit of DAG-AS on SAT03-16-INDU is due to its failure to capture the causal structure in this dataset. In cases where complex causal relationships cannot be learned due to limited training data, focusing on correlations may yield better performance.

Overall, this ablation study strongly validates the importance of the causal learning module and the DAG design in enhancing DAG-AS's performance in algorithm selection tasks. The causal graph helps better model the relationships between problem features and algorithm features, and the DAG constraint, particularly in complex tasks, helps prevent overfitting to causal information, thereby improving the accuracy of algorithm selection.

### D.4. Robustness Against Distribution Shift

In this experiment, we aimed to evaluate the robustness of DAG-AS under different types of distribution shifts and compare its performance degradation against other benchmark algorithms. The experiment focuses on observing how DAG-AS and its competitors handle performance loss when there is a shift in distribution. We designed three types of distribution shifts: (1) Shift on Problem Distribution: We selected 20% of the problem features and assigned higher sampling weights to certain feature values during the training sample selection process, while keeping the feature sampling random in the test samples.

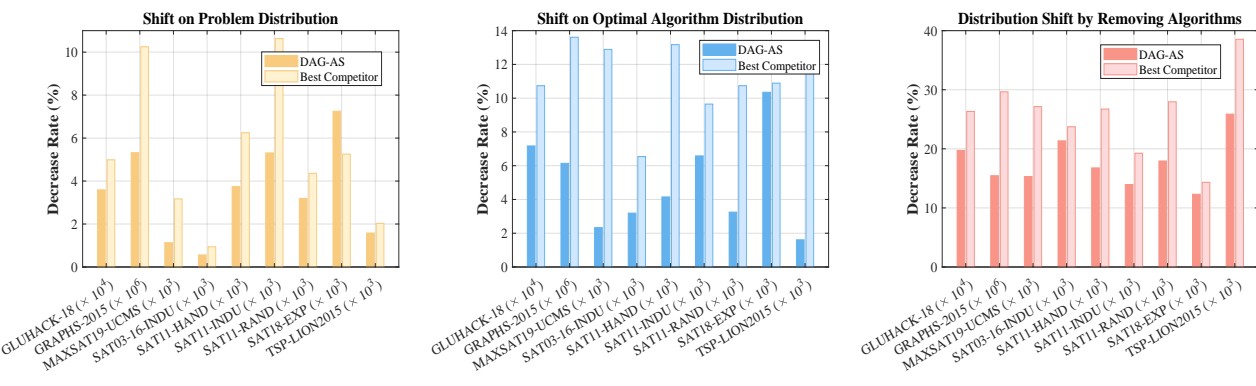

Figure 6. The generalization performance in the case with distribution shift.

(2) Shift on Optimal Algorithm Distribution: We intervened in the marginal distribution of the optimal algorithms during the training sample collection, giving higher sampling weights to samples where certain candidate algorithms achieve optimal performance, while maintaining random sampling in the test samples. (3) Distribution Shift by Removing Algorithms: We directly removed several candidate algorithms from the training data, while retaining them in the test data. To ensure fair comparison, we choose the 9 ASlib benchmarks, where the algorithm features are uniformly provided. Since all algorithms experience performance loss under distribution shift, we present the proportion of performance loss for each competitor under each distribution shift, as shown in Figure 6. For the baseline comparison, we selected the algorithm with the lowest performance loss for each dataset. The results of the three types of distribution shifts are as presented in Figure 6.

Across all three types of distribution shifts, DAG-AS consistently outperformed the best benchmark algorithms, although the magnitude of the performance advantage varied depending on the type of shift. In general, the causal learning module of DAG-AS enabled it to demonstrate strong resilience to these shifts. However, DAG-AS exhibited a particularly notable advantage when dealing with algorithm distribution shifts, which emphasizes its ability to leverage algorithm feature information more effectively than other methods.

Specifically, for the problem distribution shift, DAG-AS showed a competitive edge across most datasets, with lower performance loss compared to benchmark algorithms. This advantage stems from DAG-AS's ability to model the causal relationships between problem features and algorithm features, allowing it to remain robust even when the problem distribution is biased. However, DAG-AS showed a slight disadvantage on the SAT18-EXP dataset, where its performance loss was higher than the benchmark algorithm, indicating that its resilience to problem feature shifts may not be uniformly strong across all datasets. In contrast, the shift on optimal algorithm distribution demonstrated the greatest strength of DAG-AS. These results highlight DAG-AS's superior ability to handle biases in the optimal algorithm distribution, as it can more effectively utilize algorithm feature information to mitigate the impact of skewed training samples. This highlights DAG-AS's ability to infer the relationships between algorithms and problem features, thus avoiding overfitting to biased algorithm distributions. Finally, for the distribution shift by removing algorithms, all models faced substantial performance degradation due to the removal of candidate algorithms during training. Nonetheless, DAG-AS still demonstrated superior robustness compared to the benchmark algorithms. The causal inference module allowed DAG-AS to better infer the likely performance of missing algorithms by relying on the remaining algorithm features, thus maintaining competitive performance during testing when the removed algorithms were reintroduced.

### D.5. Demonstration of the Model-level Explainability

In the algorithm selection scenario, obtaining a causal graph that reflects the relationships between problem features and algorithm features not only helps DAG-AS achieve more accurate results but also enables us to derive many meaningful conclusions through causal interpretation. By analyzing the learned causal relationships, we can gain deeper insights into the underlying interactions between features and improve the decision-making process in algorithm selection.

**Feature Importance in the Causal Graph:** The first part of this experiment focuses on analyzing the importance of all

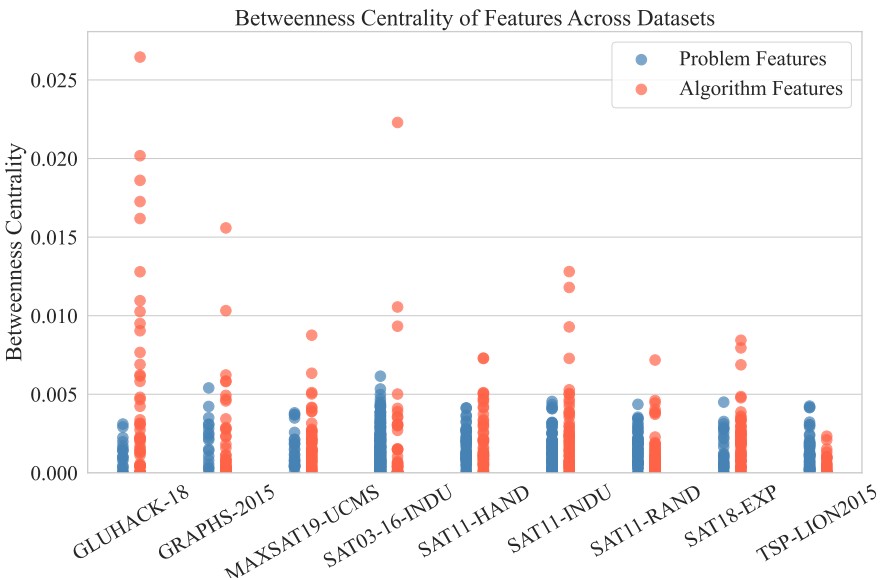

*Figure 7.* Betweenness centrality of features.

problem and algorithm features in the causal graph across different datasets. We measure the betweenness centrality of each feature in the DAG, as shown in Figure 7. From the results, it is evident that high betweenness centrality values are predominantly associated with algorithm features, while the betweenness centrality of problem features remains relatively similar across different datasets. This observation highlights the importance of algorithm features in the context of algorithm selection. Algorithm features provide more informative cues than mere algorithm categories and often serve as outcome variables in causal relationships. By interpreting these relationships, the model can establish a more compact and interpretable representation of the interactions between problem features, leading to better overall performance in algorithm selection.

*Table 4.* The correspondence between the IDs and names of algorithm features. All names are consistent with those in the ASlib benchmark. Algorithm features not included in the table have no edges in the causal graph across all data and are considered invalid features.

| ID | Name | ID | Name | ID | Name | ID | Name |
|----|------|----|------|----|------|----|------|
| 0 | Lines..Average. | 1 | Lines..Total. | 2 | Size..Average. | 3 | Size..Total. |
| 4 | Number.of.files | 5 | Cyclomatic..Average. | 6 | Cyclomatic..Total. | 7 | Max.Indent..Average. |
| 8 | Max.Indent..Total. | 9 | nb_nodes | 10 | nb_edges | 11 | degree_max |
| 12 | degree_mean | 13 | degree_variance | 14 | degree_entropy | 15 | transitivity |
| 16 | clustering_mean | 17 | clustering_variance | 18 | paths_max | 19 | path_mean |
| 20 | path_variance | 21 | path_entropy | 22 | Stmt | 23 | Type |
| 24 | Decl | 25 | Attribute | 26 | Operator | 27 | Literal |
| 28 | edge_ss | 29 | edge_st | 30 | edge_sd | 31 | edge_sa |
| 32 | edge_so | 33 | edge_sl | 34 | edge_ts | 35 | edge_tt |
| 36 | edge_td | 37 | edge_ds | 38 | edge_dt | 39 | edge_dd |
| 40 | edge_as | 41 | edge_at | 42 | edge_ad | 43 | edge_os |
| 44 | edge_od | 45 | edge_oo | 46 | edge_ls | 47 | edge_ld |
| 48 | edge_lo | 49 | op_short | 50 | op_int | 51 | op_long |
| 52 | op_long_long | 53 | op_float | 54 | op_double | 55 | op_bit |

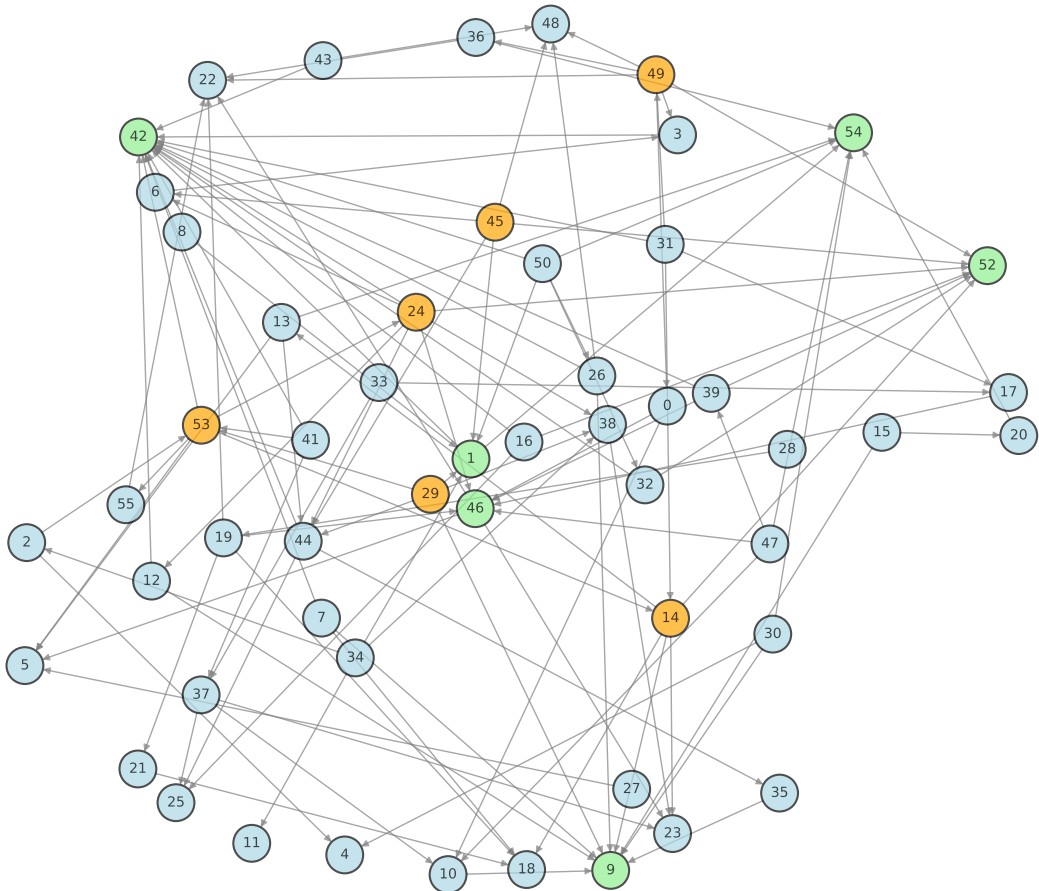

*Figure 8.* Causality between algorithm features. The correspondence between **the IDs and names** of algorithm features are provided in **Table 4**.

**Causal Relationships Among Algorithm Features:** Next, we investigate the causal relationships between algorithm features across nine datasets. All valid algorithm features, along with their names and IDs, are recorded in Table 4. The remaining features are considered invalid, as they have no edges in the causal graph across all data. We averaged the causal graphs obtained from all datasets and visualized the most confident causal links in Figure 8. In this figure, nodes with the highest 10% in-degree and out-degree are highlighted in green and orange, respectively. The causal graphs reveal widespread interdependencies among algorithm features within the ASlib benchmark, though these relationships tend to be sparse. This sparsity suggests a balance between the predictive power and redundancy of the algorithm feature data. As problem features vary across datasets, the importance of different algorithm features shifts accordingly. By modeling this relationship through causal graphs, the model can abstract the most predictive algorithm features and identify key variable relationships, enhancing its performance in algorithm selection.

**Impact of Problem Features on Algorithm Features:** To further analyze the influence of problem features on algorithm features, we focused on the cause sets of algorithm features in the causal graph of each dataset. Figure 9 illustrates the proportion of problem features directly influencing algorithm features, relative to the total number of problem features in each dataset. Since both problem and algorithm features in ASlib are manually designed, and the design of problem features varies across datasets, this result allows us to evaluate the rationality of feature design by observing the density of causal relationships in each dataset. We found that datasets with denser causal relationships, such as MAXSAT19-UCMS, SAT11-HAND, SAT11-INDU, and SAT18-EXP, also exhibited a larger performance advantage for DAG-AS. This indicates that causal relationships are prevalent between problem and algorithm features in these datasets, and leveraging these causal relationships allows DAG-AS to better exploit the potential of algorithm features. In contrast, in datasets with sparser causal relationships, such as SAT03-16-INDU and GLUHACK-18, the advantage of DAG-AS was diminished, and in some cases, DAG-AS even underperformed compared to benchmark algorithms. In these datasets, the causal learning module was only

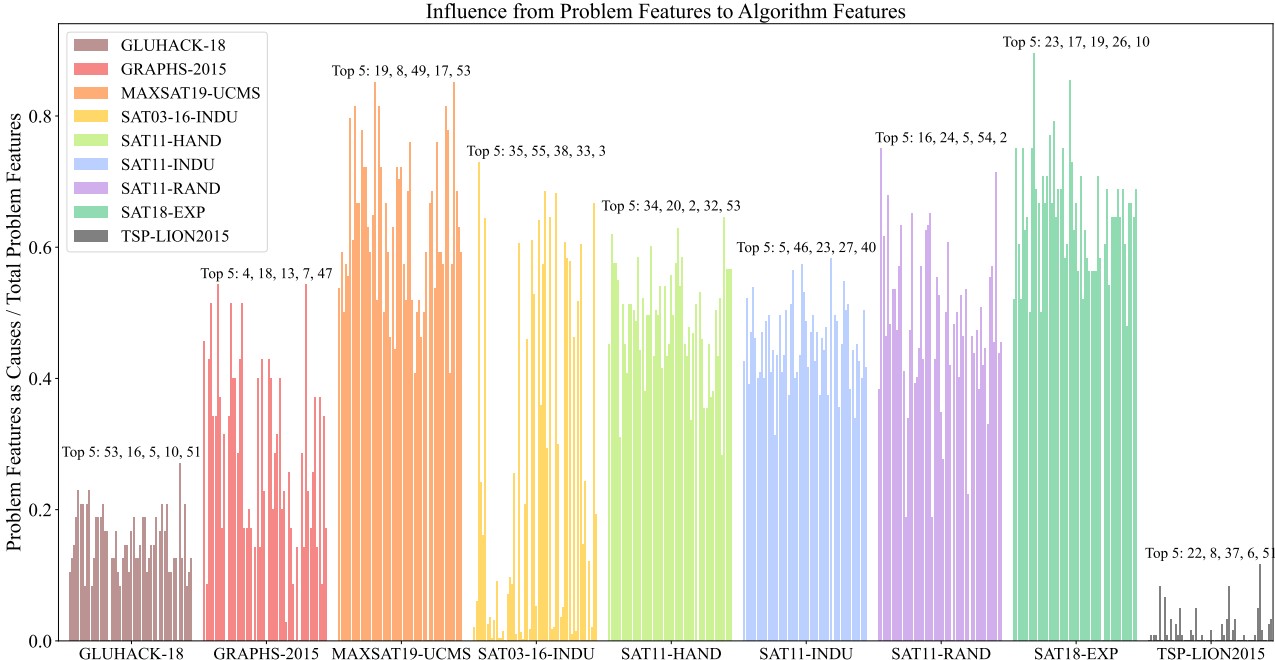

*Figure 9.* Influence from problem features to algorithm features.

able to identify a limited number of causal relationships, resulting in models that lacked robustness. This outcome suggests that we can estimate the performance of DAG-AS in specific scenarios by analyzing the learned causal graphs.

One notable exception is the TSP-LION2015 dataset, where DAG-AS still achieved the best performance despite the causal relationships being relatively sparse. This can be attributed to the nature of the dataset: TSP-LION2015 is a relatively simple benchmark with a large number of training samples and only four candidate algorithms. The abundance of training data enabled DAG-AS to retain the most critical causal relationships while filtering out redundant ones, allowing it to maintain a significant performance advantage. This case demonstrates that in simpler scenarios with sufficient training data, DAG-AS can still excel even when causal relationships are sparse, as it can effectively focus on the most influential relationships.

### D.6. Demonstration of the Instance-level Explainability

In the algorithm selection modeling process, DAG-AS generates a causal graph that enables the derivation of counterfactual explanations through interventions based on this graph. This section focuses on a demonstrative experiment conducted on the GRAPHS-2015 dataset to showcase this explainability.

We solve the optimization problem outlined in Eq (17) in the main text to determine the minimal intervention values that can alter DAG-AS's selection results. To streamline the problem and narrow the scope, we impose specific constraints: (1) a maximum of 20% of the features can be intervened upon; (2) the intervention's magnitude must be less than 10%, with the modified values remaining within the valid range of their respective variables. Under these constraints, we identify 79 instances where the algorithm selection result can be shifted from selecting one algorithm to another, as illustrated in Figure 10. The first heatmap on the left depicts the total intervention magnitude for each sample, with darker colors indicating larger interventions. The second heatmap on the right shows the intervention magnitudes for each feature across samples, where white indicates no intervention, and color gradients from light yellow to deep blue or red reflect small to large positive or negative interventions, respectively. The arrows between the heatmaps indicate decision changes by DAG-AS, such as "$a \rightarrow b$," which denotes a transition from selecting algorithm a to algorithm b.

**Case Study of Instance 1846:** Taking Instance 1846 as an example, applying interventions of -0.00796, -0.00339, -0.07072, -0.00342, 0.00252, -0.00003, and -0.05042 to features 10, 11, 13, 14, 16, 26, and 28 respectively alters its optimal algorithm choice from candidate algorithm 3 to candidate algorithm 1. Conversely, applying interventions of 0.01116, 0.00685, 0.07201, 0.02622, 0.19386, -0.00242, and -0.15107 to features 2, 3, 5, 6, 8, 19, and 21 changes the choice from candidate

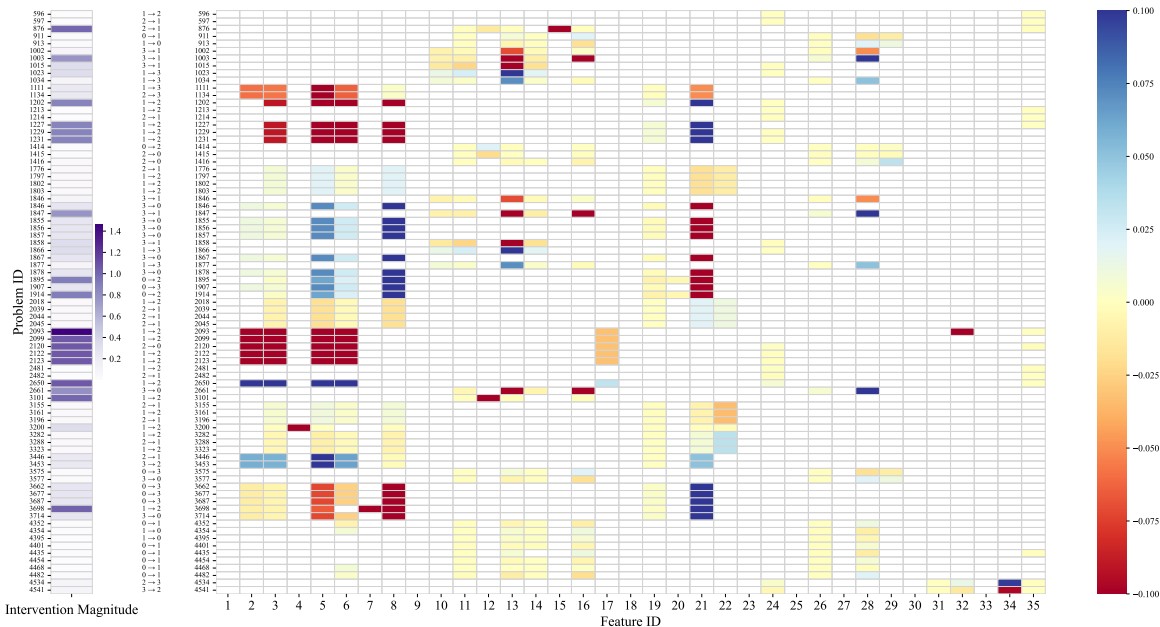

*Figure 10.* Demonstration of counterfactual explainability on GRAPHS-2015 dataset. The correspondence between the IDs and names of algorithm features are provided in Table 4. Algorithm IDs 0, 1, 2, and 3 correspond to glasgow2, lad, supplementallad, and vf2 in the GRAPHS-2015 dataset, respectively.

algorithm 3 to candidate algorithm 0.

**Insights from the Heatmaps:** The heatmaps indicate that small interventions on certain variables can effectively prompt DAG-AS to make different selection choices. Features 11, 13, 14, 16, 19, and 26 exhibit high sensitivity to perturbations, requiring minimal changes to influence DAG-AS's decision-making. Although features 2, 3, 5, 6, and 8 are less sensitive, their significant interventions can still lead to different outcomes. Overall, only 1.36% of all samples can shift their selection results under the defined limits and range of interventions. Notably, aside from Instance 1846, which exhibited two possible direction shifts, all other samples transitioned towards a single candidate algorithm. This underscores the robustness of DAG-AS's decision-making process.

This analysis highlights DAG-AS's capability to offer interpretable insights through causal graph-based counterfactual reasoning. By strategically applying minimal interventions, we can gain a deeper understanding of how specific features influence algorithm selection, thereby enhancing the transparency and reliability of DAG-AS in practical applications.

