# OpenReview forum: "Towards Robustness and Explainability of Automatic Algorithm Selection"
_ICML.cc/2025/Conference — ICML 2025 spotlightposter_

### Official Review · Reviewer_oy94 · 2025-02-25

**Overall Recommendation:** 4

**Summary:**

The paper focuses on the automatic algorithm selection problem. It aims to improve the explainability and robustness of algorithm selection. Main contributions include:

(1) The most significant innovation of this paper is that it changes the modeling approach of the algorithm selection task. Traditionally, models predict which algorithm to choose based on problem features and algorithm features. However, this paper's approach predicts algorithm features based on problem features. In other words, it focuses on "What characteristics an algorithm needs to solve a problem with specific feature values".

(2) Causal Modeling with DAG: The paper introduces a causal framework using DAGs to describe the underlying mechanism of algorithm selection.

(3) Model Framework: The DAG-AS model is based on a neural network framework incorporating causal learning principles. The model reconstructs algorithm features, and the final algorithm selection is made by comparing the oringinal features of candidate algorithm and reconstructed features of the optimal algorithm.

**Claims And Evidence:**

Overall, the claims made in the submission are supported by clear and convincing evidence, including:
(1) DAG-AS's performance superiority
(2) Ablation study
(3) DAG-AS's robustness against distribution shifts
(4) The model-level explainability of DAG-AS
(5) The instance-level explainability of DAG-AS

**Essential References Not Discussed:**

It is recommended to add a discussion on explainability in the Background section.

**Experimental Designs Or Analyses:**

The experimental designs and analyses in the paper, including performance comparison, ablation study, robustness tests, and explainability experiments, are generally sound and valid for validating the DAG-AS model.

**Methods And Evaluation Criteria:**

The evaluation is conducted based on ASlib Benchmark and PAR10 Score, whose use aligns with the algorithm selection community's established practices.

**Other Comments Or Suggestions:**

N/A

**Other Strengths And Weaknesses:**

Strengths:

1. The paper introduces causal learning into the field of algorithm selection, offering a brand-new perspective to address the limitations of traditional methods. The DAG-AS model constructs a causal graph to clarify the causal relationships between problem features and algorithm features. This innovative modeling approach is groundbreaking in algorithm selection research.

2. The paper makes significant contributions in terms of explainability. The model-level explanation can visually display the dependency relationships among variables by analyzing the DAG structure, helping researchers understand the model's decision-making process. The instance-level explanation uses counterfactual interventions to accurately identify the key features that affect algorithm selection, providing actionable explanations for practical applications and enhancing the credibility and practicality of the model.

3. The experimental design is comprehensive and rigorous. The research covers various aspects such as performance comparison, ablation studies, robustness testing, and explainability verification, providing strong support for the viewpoints in the paper.

Weaknesses:

1. How to obtain algorithm features is a prerequisite for this study. The method in this paper assumes that algorithm features are known, but there are scenarios where algorithm features are not provided. Even in ASlib, only 10 datasets contain algorithm features. I suggest that the authors clearly introduce how to obtain algorithm features in the paper, which will improve the practicality of the method.

2. The Discussion in P7 is interesting. I think that after making significant changes to the modeling paradigm of algorithm selection in this paper, one of the key advantages is that the algorithm selection model can, in turn, promote the improvement of algorithms. Unfortunately, this paper does not discuss this content in detail. Therefore, I suggest either enriching this part of the content or including it in future work.

3. A minor issue: The symbol "a" in line 371 should be explained.

**Questions For Authors:**

1. The paper achieves multi-level explainability in algorithm selection, which represents the new value emerging from the application of causal concepts and recommendation idea into the algorithm selection field. Are the goals of this explainability consistent with those of the classic SHAP analysis? Or does the connotation of the explainability in this paper cover and go beyond what the SHAP analysis encompasses?

2. In instance-level explanations, the algorithm selection is explained by finding the minimum perturbations through counterfactual interventions. However, in practical operations, how can we determine the appropriate perturbation range and step size to ensure that effective explanations can be obtained? In the experiments of Figure 10, the authors achieved this by imposing certain constraints. But the optimization problem modeled in Eq. (15) does not involve these constraints. Is it necessary to make the form of the optimization problem in Eq. (15) consistent with that in the experiments?

3. How to obtain algorithm features is a prerequisite for this study. The method in this paper assumes that algorithm features are known, but there are scenarios where algorithm features are not provided. Even in ASlib, only 10 datasets contain algorithm features. I suggest that the authors clearly introduce how to obtain algorithm features in the paper, which will improve the practicality of the method.

4. In the ablation study, is the construction of the directed cyclic graph still based on the two assumptions of this paper? Why does the method using DAG perform worse than the method without considering causality in some scenarios?

**Relation To Broader Scientific Literature:**

Researchers in the AutoML and algorithm selection community will benefit from this study. The paper revolutionizes the modeling approach in algorithm selection, offering new inspiration for subsequent research, especially regarding explainability and robustness. However, while the paper uses techniques from causal learning and recommendation systems, its inspiration for these two fields is limited.

Overall, the primary audience for this paper is in the AutoML field, especially algorithm selection researchers. The findings and methods presented can be directly applied or adapted to improve algorithm selection algorithms, making it a valuable contribution to this specific research community.

**Theoretical Claims:**

The paper presents Theorem 2.3 to support the use of causal models in the algorithm selection task. The proof appears to be correct in its logical structure. It first shows the existence of an algorithm feature whose parent set is composed of problem features through a proof by contradiction. Then, it uses topological ordering and mathematical induction to construct functions for each algorithm feature's conditional distribution.

---

> ### Author Rebuttal · Authors · 2025-03-31
>
> > Weakness 1 / Q3
>
> Thanks for your insightful comments. In practical scenarios, informative algorithm features are not always readily available, just as pointed out by Reviewer nswt. In this study, we recognize that the availability of algorithm features has a vital impact on the model's performance. Therefore, we will add an assumption and discussion about the informativeness of algorithm features in Section 2:
>
> **Assumption 3**: Informative algorithm features are available for the algorithm selection task.
>
> In the context of algorithm selection, the availability of informative algorithm features is a crucial yet often overlooked aspect. This assumption implies that we can obtain algorithm-related characteristics that carry meaningful information for differentiating the performance of different algorithms on various problem instances. This assumption serves as a fundamental basis for DAG-AS. Currently, there are several established algorithm representation methods, such as hyperparameters, model structure information, code-related statistical features, AST features, and code features extracted by LLMs. These features provided valuable insights into the algorithms' properties and significantly contributed to the performance of DAG-AS. However, we are also aware that in some cases, obtaining such informative features can be challenging. For example, code-related features cannot be obtained in the closed-source scenario.
>
> > Weakness 2
>
> We sincerely appreciate your valuable suggestions. Intervening on algorithm features indeed holds the potential to achieve broader goals beyond the algorithm selection task itself. In real-world scenarios, the most suitable algorithm may not always be present in the candidate set, or may not even have been designed yet. By examining $P(S=1 \mid \textbf{PF},do(\textbf{AF}=\textbf{a}+\delta_{\textbf{AF}}))$, we can explore how to adjust the optimal algorithms within the candidate set based on $\delta_{\textbf{AF}}$ for a given problem. This intervention can assist us in two ways:
> 1. It helps in searching for more suitable algorithms within or beyond the existing candidate set;
> 2. It generates valuable suggestions to aid in the automated algorithm design.
>
> However, in the context of ASlib, due to the high level of abstraction of the algorithm features and the fact that many candidate algorithms are closed-source, we are unable to optimize these candidate algorithms according to the results of algorithm feature interventions. As a result, empirical study in this aspect is currently not feasible. In the future, such research could potentially be carried out in scenarios where the physical meaning of features is more explicit. For example, in the selection of deep-learning models, the hyperparameters and architecture features of the models can be used as algorithm features. Interventions on these features can help users to further improve existing models or generate new ones based on the candidate algorithm set.
>
> Regarding the generation of new algorithms, we already have some well-developed ideas. Specifically, by combining the DAG learned by DAG-AS and the results of algorithm feature interventions, we can generate suggestions for improving candidate algorithms. These suggestions, along with the code of the candidate algorithms, can then be submitted to a LLM. Leveraging the code-related ability of the LLM, we can achieve the automated optimization and design of algorithms. We will incorporate the discussion into the final version to further enhance the depth of our research.
>
> > Q1
>
> The goal of classic SHAP analysis is to explain the prediction results of a model by calculating the contribution of each feature to the prediction. It focuses on quantifying the impact of individual features on the output within the model's framework. In contrast, our paper aims to achieve multi-level explainability in algorithm selection. It not only considers the influence of features on the selection result but also delves into the causal relations between problem features and algorithm features, as well as the overall causal structure. This helps users understand not only which features are important but also how they interact and influence the algorithm selection process in a causal sense.
>
> > Q2
>
> We appreciate your astute observation. We indeed introduced additional constraints to simplify the search process for determining the appropriate perturbation range and step size in instance-level explanations. As detailed in our last response to the Reviewer BWG1, we have proposed some methods to assist in solving the optimization problem. Due to space limitations, we cannot elaborate on these here. Moreover, it is unnecessary to adjust Eq.15 since this equation represents the original form of the optimization problem.
>
> > Q4
>
> The construction of the directed cyclic graph still based on the two assumptions. As for performance degradation on SAT03-16-INDU, please read the 5th response to Reviewer nswt.

---

> > ### Comment · Reviewer_oy94 · 2025-04-08
> >
> > I have read the comments by the authors and have no further questions.

---

### Official Review · Reviewer_nswt · 2025-03-03

**Overall Recommendation:** 5

**Summary:**

The paper introduces a new approach to algorithm selection using directed acyclic graphs (DAGs) and causal relations. The approach focuses on modeling the causal relations between problem features and algorithm features. The authors argue that this method not only improves the accuracy of algorithm selection but also enhances the model's robustness against distribution shifts and provides multi-level explainability. This approach allows the model to learn the strong feature representations for the most suitable algorithm given a specific problem. The authors of the given paper also propose a counterfactual explanation method to understand how minimal changes in problem features can lead to different algorithm selection outcomes, improving the interpretability of algorithm selection. The authors demonstrate the effectiveness of their approach through experiments on the ASlib benchmark, showing superior performance compared to many traditional algorithm selection methods.

**Claims And Evidence:**

I strongly appreciate the proposed approach of applying causal relationship learning with algorithm selection. I agree with the authors that this is in fact a very novel approach for the domain of algorithm selection and makes major contributions to improving the performance of algorithm selection systems.

The empirical evidence of the paper is strong since the authors compare against established algorithm selection approaches on an established algorithm selection benchmark library (ASlib). However, I’m concerned about missing state-of-the-art approaches for algorithm selection. For example, in the Open Algorithm Selection Challenge ASAP won the competition. Other strong and established systems such as AutoFolio are also not considered. ISAC, SATzilla11 and SNNAP are very old approaches and not considered SOTA. (It is less clear for SUNNY since it would depend on the version of SUNNY.)

I very much like the idea of counterfactual interpretations of algorithm selection. However, it is not fully clear to me whether this depends on the DAG approach proposed by the authors or whether this is completely orthogonal to the rest of the paper. To my understanding, counterfactual interpretation can also be derived on correlation-based approaches.

I fully agree with the claim of robustness of the DAG approach and the conducted experiments.

**Essential References Not Discussed:**

None

**Experimental Designs Or Analyses:**

The experimental design was overall convincing (with the few exceptions I mentioned above already). I also strongly appreciated the ablation and robustness study showing the strengths and advantages of the approaches and its components.

I was missing a bit more details on when the approach will perform well and why it failed to do so on two ASlib scenarios. The authors briefly mention aspects such as the amount of training data; however, for example, SAT03-16-INDU has a lot more training data than SAT11-RAND; nevertheless, DAG-AS performed very well on SAT11-RAND but not on SAT03-16-INDU. Later on, the authors argued with the sparsity of the DAG; however, also the additional experiments in the appendix (e.g., Figure 9) have not really helped me to get a good grasp of when DAG-AS fails and when not.

As discussed before, I’m least convinced by the interpretation advantage of DAG-AS. I agree that having a DAG is nice in practice, but DAGs as shown in Figure 8 are not helpful in understanding the underlying modeling problem and causal relationship. Although I am quite familiar with the ASlib scenarios, I was not able to get any good insight from Figure 8. Also as said before, I like the idea of applying counter-factual interpretations and, indeed, a DAG makes this a lot easier, but in principle, methods such as LIME could also be used similarly.

**Methods And Evaluation Criteria:**

The proposed method is very compelling and nicely motivated. The formal derivation is good and with sufficient background knowledge also fairly understandable. Some minor comments: It could be clearer which modeling assumptions are specific to algorithm selection and which ones are quite established modeling approaches for causal learning. Furthermore, I dislike that I had to jump to the appendix to read up on parts of the notation – without it, someone not being fully familiar with causal learning notation is lost in the main paper.

The evaluation on ASlib with PAR10 follows the common best practices of the community and thus it is very appropriate. However, it is a bit unfortunate that only some of the benchmark scenarios have algorithm features and some others don’t have them s.t. it is not quite clear whether the benchmarks could (unintentionally) biased.

Furthermore, I was missing any kind of baseline that makes use of algorithm features, as the ones cited by Tornede et al. As far as I know, none of the baselines use algorithm features, but only instance features.

**Other Comments Or Suggestions:**

None

**Other Strengths And Weaknesses:**

I see a lot of potential in this paper because it provides a completely new direction for the algorithm selection community. It is the first very substantial progress in algorithm selection I have seen in recent years.

I was hoping to see more discussion on the intervention of algorithm features in this paper since causal reasoning would make this much more interesting. I wonder whether the authors were thinking about modifying instances or generating completely new instances – both are research directions already proposed (independent of causal reasoning) in the algorithm selection community.

Furthermore, informative algorithm features are not always easily available. Also, the computation graph features proposed by Pulatov et al. are not as trivial and it is unclear whether they are informative enough. I would love to have at least a short paragraph discussing the very important assumption on the availability of informative algorithm features.

**Questions For Authors:**

I have a bit of concern regarding your weighted loss function which seems to be fairly complex. I fully understand and agree with the individual components. But how have you managed to find the weights for your loss functions and on which benchmarks have you done this?

**Relation To Broader Scientific Literature:**

The related work is overall well covered. It seems to be a bit biased towards the traditional approaches in algorithm selection and discusses less modern approaches based on deep learning without explicit feature representations.

**Theoretical Claims:**

Theorem 2.3 and all the other theoretical derivations made sense to me intuitively, as I am someone with strong expertise in algorithm selection and less in causal learning. I have not read Proof B (in the Appendix) in detail.

---

> ### Author Rebuttal · Authors · 2025-03-31
>
> > Experiments: Missing SOTA approaches. None of the baselines use algorithm features.
>
> Thanks for your valuable suggestions. We acknowledge that our experiments lacked some SOTA comparison methods. In response, we have now incorporated ASAP and AutoFolio into our experiments.
>
> Regarding the absence of methods based on algorithm features in our original study, there are two main reasons: 1) these methods mainly differ in feature types rather than model design, and 2) in some scenarios of ASlib, it was difficult to obtain the source code, which made it challenging to extract algorithm features, such as features derived from code or hyperparameters.
>
> According to your suggestion, we selected the method proposed by Pulatov et al and AS-LLM as additional comparison baselines. Due to space constraints, we have plotted the experimental results in a figure and presented them at the anonymous link https://imgur.com/a/QmDoUeA. It should be noted that AS-LLM can only be evaluated in scenarios where the source code is available. From the experimental results, it can be clearly seen that DAG-AS still maintains the best performance in most scenarios.
>
> > Whether counterfactual interpretations depends on the DAG-AS
>
> Counterfactual interpretations are indeed closely related to DAG-AS. As described in Page 7, the three steps of counterfactual calculation all rely on the causal model learned by DAG-AS:
> 1. Abduction step needs to use the structural equations in DAG-AS to infer the exogenous variables;
> 2. Intervention step uses do-operator to modify the causal graph learned by DAG-AS.
> 3. Prediction step calculates the results based on the modified causal graph.
>
> In summary, while counterfactual interpretations can be conceptually applied in other settings, in our work, the DAG-AS framework provides the necessary foundation for a systematic and accurate implementation of counterfactual analysis in algorithm selection.
>
> > Which modeling assumptions are specific to algorithm selection and which ones are quite established modeling approaches for causal learning.
>
> The two assumptions in the paper are both proposed based on the characteristics of the algorithm selection task, mainly to simplify the complexity of the causal graph. There are some assumptions in causal learning itself, such as causal sufficiency, which are not included in this paper.
>
> > Someone not being fully familiar with causal learning notation is lost in the main paper.
>
> We sincerely apologize for any confusion caused to readers. We will make adjustments and add a notation tabel into the final version.
>
> > When DAG-AS fails and when not
>
> We think that DAG-AS may not perform well under 3 cases:
> 1. When the features themselves lack informativeness or when the correlation between problem features and algorithm features is extremely weak.
> 2. When the data violates the causal sufficiency assumption. This means that there are confounding factors that are not included in the feature set.
> 3. When the causal relationships within the dataset are overly complex.
>
> However, it should be noted that cases 2 and 3 can potentially be addressed by improving DAG-AS. In the causal learning field, there are numerous specialized models designed to handle complex causal relations, such as those dealing with multivariable or nonlinear causality. Additionally, there are studies focused on identifying causality in scenarios where the causal sufficiency assumption is violated. To enhance the performance of DAG-AS, more advanced causal learning models can be adopted to replace the Eq(8,9).
>
> > I’m least convinced by the interpretation advantage of DAG-AS.
>
> We sincerely appreciate your criticism. We acknowledge that the interpretability aspect in the context of ASlib may not be as immediately intuitive. The reason is that the algorithm features used in ASlib are AST features, which are highly abstract in terms of their physical meaning. However, the interpretability can be extremely valuable in scenarios where the physical meaning of features is explicit. For example, when considering the architecture features or hyperparameters of deep-learning models as algorithm features. In such cases, if the causal graph reveals which problem features influence the number of neurons in specific layers, this interpretability can significantly assist users in understanding the model design. It also provides great utility for experts when debugging the DAG-AS model.
>
> **Due to space limitations, the following issues are addressed in the response of other Reviewers:**
>
> > Paragraph discussing the assumption on the availability of informative algorithm features.
>
> Please refer to the 1st response of Reviewer oy94 (Weakness1)
>
> > More discussion on the intervention of algorithm features
>
> Please refer to the 2rd response of Reviewer oy94 (Weakness2)
>
> > Weights for loss function
>
> Please refer to the 2rd response of Reviewer BWG1 (Weakness2)

---

> > ### Comment · Reviewer_nswt · 2025-04-02
> >
> > Many thanks for the reply, explanations and new results. Very much appreciated. I increased my score accordingly since I believe the few weaknesses do not matter in view of this important and very novel contribution to algorithm selection.
> >
> > However, I would like to slightly disagree with your wording in the rebuttal:
> >
> > > physical meaning of features is explicit
> >
> > For algorithms, there is nothing like a physical meaning since they are always abstract concepts ;-)
> > If you would like to add this line of arguments to your paper, you will find better wording for that.

---

> > > ### Author Response · Authors · 2025-04-03
> > >
> > > We sincerely appreciate your strong support for our paper and your insightful review comments. Your feedback has been extremely valuable to us.
> > >
> > > We acknowledge that stating there is a "physical meaning" for algorithm features was inaccurate. What we actually intended to convey is that when algorithm features possess an **intuitive or tangible meaning that is easily comprehensible to humans**, the explanations provided by the causal graph, as well as the interventions on these algorithm features, can significantly aid humans in understanding the algorithm selection mechanism and improving candidate algorithms.
> > >
> > > We will use more precise wording in the final version of our paper to avoid such misunderstandings. Once again, thank you for your attention to detail and for helping us improve the quality of our work.

---

### Official Review · Reviewer_BWG1 · 2025-03-11

**Overall Recommendation:** 3

**Summary:**

The paper argues that current approaches to automatic algorithm selection are mainly based on the correlation between algorithm performance and problem meta-features, which are susceptible to data bias and distributional variations, and lack robustness. The thesis proposes to use DAG to represent the causal relationship between problem features and algorithm features, to model the algorithm selection process, and to provide model-level and instance-level interpretability through counterfactual computation. In ASlib benchmark tests, DAG-AS outperforms existing methods in terms of robustness and interpretability.

**Claims And Evidence:**

Yes.

**Essential References Not Discussed:**

The reference is sufficient.

**Experimental Designs Or Analyses:**

The authors‘ experimental setup seems reasonable to me, but in the experimental design for distributional shifts in Appendix E.4, I am puzzled by the authors’ simulation of ‘distributional shifts of the optimal algorithm’, the distribution of a particular optimal algorithm, because for a well-designed algorithm, its characteristics should be fixed, and for a defined problem characteristic, the optimal algorithm should also be defined. Intuitively, the optimal algorithm should also be fixed.

**Methods And Evaluation Criteria:**

Yes, the approach proposed by the authors models the relationship between problem features and algorithm features from a causal perspective and provides interpretability.

**Other Comments Or Suggestions:**

1.	It is recommended that the authors provide further details on the practical applications of this direction of automatic algorithmic selection as well as on the issue of efficiency to help the reader better understand the importance of this direction.

2.	Even though the questions are different, I still recommend that the authors read [1], whose modelling of causality and discussion of it is very worthwhile for the authors to learn from.

[1] CausPref: Causal Preference Learning for Out-of-Distribution Recommendation

**Other Strengths And Weaknesses:**

Strengths:

1.	Existing algorithm selection methods mainly rely on empirical correlation, the paper is the first to use DAG to describe the relationship between algorithmic features and problem features based on causal modelling, thus enhancing the robustness and interpretability of the model.

2.	The paper provides an actionable explanation for algorithmic recommendations by analysing how problem characteristics affect algorithmic choices through minimal intervention in an instance-level explanation.

3.	The paper uses multiple causal diagrams (DAG structure), heatmap, and performance comparison table to illustrate the advantages of DAG-AS, making the experimental results intuitive and easy to understand.

Weakness:

1.	The question of the reasonableness of the distributional bias experiment (Appendix E.4(2)) setup, i.e., the reasonableness of modifying the optimal algorithmic distribution?

2.	The authors do not have corresponding hyperparameters to balance the three aspects of constraints on causality, constraints on feature representation, and constraints on the DAG structure, and do not see this aspect discussed in the experiments. I think this is necessary.

**Questions For Authors:**

1.	Question about Eq. 2:

P(S = 1|PF, AF) seems to be the existing modelling approach, the authors proved through Bayes‘ Theorem the positive relationship between the modelling approach taken by the authors, P(AF|PF, S = 1), and P(S = 1|PF, AF), in my understanding, the advantage of causal modelling over correlational modelling lies in its identification of confounding factors and consideration of the variables’ causal relationships, why did the authors make this clarification? The advantages of modelling causality over modelling correlation seem obvious to me.

2.	Is there an efficiency problem with exploring interpretability through counterfactual methods? Because it seems that the do operation needs to iterate through all PFs?Do the authors have anything relevant to say about this?

**Relation To Broader Scientific Literature:**

Previous studies modelled the joint distribution of PF and AF, and the authors concluded that such an approach is sensitive to changes in the distribution and has poor robustness, and the authors proposed modelling P(AF|PF, S = 1) to consider automatic algorithmic choices from a causal perspective.

**Theoretical Claims:**

The author's proof seems reasonable to me.

---

> ### Author Rebuttal · Authors · 2025-03-31
>
> > Weakness 1
>
> Thanks for your valuable comments. It's right that "for a defined problem characteristic, the optimal algorithm should also be defined", which implies that $P(AF|PF)$ remains constant. In Appendix E.4, our intention was not to change $P(AF|PF)$, but rather to manipulate the marginal distributions $P(AF)$ and $P(PF)$. The “Shift on Optimal Algorithm Distribution” was achieved through a deliberate selection process of the training and test data. Specifically, any candidate algorithm can be the optimal ones for a certain proportion of problems. We manipulate this proportion to induce a distribution shift of the optimal algorithm. For example, in the training data, algorithms A and B are the best-performing ones for 20% and 80% of the problems respectively, while in the test data, these proportions are reversed, with A being optimal for 80% and B for 20% of the problems. We construct the training and test sets by selecting problem instances from the original dataset according to these proportions. Since $P(AF|PF)$ is fixed in algorithm selection, any adjustment to either $P(AF)$ or $P(PF)$ will lead to a corresponding change in the other. Therefore, during the implementation of the “Shift on Optimal Algorithm Distribution”, the distribution of problem features also changed accordingly.
>
> > Weakness 2
>
> Thanks for your reminder. We have used hyperparameters to balance the impacts of various losses. As stated on line 278, “The overall loss function is the weighted average of the causal learning loss and algorithm selection loss.” However, we apologize for the misunderstanding caused by not presenting the final form of loss function. In the final version, we will provide the complete loss function with hyperparameters: $L = L_{\text{reconstruction}} + \alpha L_{\text{sparsity}} + \beta L_{\text{acyclicity}}+ \gamma L_{\text{selection}}$. Current experimental results were obtained under the same parameter settings, where $\beta=\gamma=1$ and $\alpha=0.0001$. Based on your suggestion, we conducted a hyperparameter analysis on SAT11-INDU. The results can be found at the anonymous link: https://imgur.com/a/b8C9l5G. In left figure, we analyzed the balance between reconstruction loss and two causal learning constraints (analyzing $\alpha$ and $\beta$), while keeping $\gamma=1$. In right figure, we analyzed the balance between the causal learning loss and the algorithm selection loss (analyzing $\gamma$), while keeping $\beta=1,\alpha=0.0001$. It can be seen that DAG-AS can maintain relatively good performance within a wide range of parameters. In the final version, we will supplement more detailed hyperparameter analysis.
>
> > Other Comments & Suggestions
>
> We sincerely appreciate your suggestions. We'll incorporate more details into the final version to highlight the significance of this area, including its practical applications in machine learning, scientific computing, and engineering optimization, as well as its superiority in enabling efficient decision-making, especially in real-time applications. In terms of CausPref, it is a typical study in which causal learning enhances robustness. The strategies of CausPref in causal preference modeling and negative sampling are worth learning. We will cite this paper in the final version and discuss the inspiration for our study.
>
> > Q1
>
> We apologize for the ambiguity in this part. Our intention was not to illustrate “The advantages of modelling causality over correlation” through Eq(2). Rather, we aimed to express the connection between the two modelling approaches, i.e., $P(S=1|PF,AF)$ and $P(AF|PF,S=1)$ are equivalent in terms of modelling in the algorithm selection task. However, $P(AF|PF,S=1)$ directly models the conditional distribution, which enables it to be more resilient against the marginal distribution shifts. After these discussions, we propose using a causal DAG to model $P(AF|PF,S=1)$.
>
> > Q2
>
> First, we would like to clarify that the process does not necessarily require iterating through all PFs. This mainly depends on the DAG learned by DAG-AS. We only need to intervene on all the parent nodes of the algorithm features. Therefore, the DAG has already reduced the dimensionality for counterfactual calculations.
>
> However, even with this reduction, a large number of problem features may still be involved in the counterfactual calculations, so improving the efficiency of this process is essential. One potential approach is to model the interpretability calculations as a reinforcement learning system, where the action space would include the features of intervention and their magnitudes, and the reward space would be defined by the changes in the algorithm selection decision. We could take several interventions as samples to train separate Action and Reward networks. This approach may require an independent research paper for in-depth discussion. Therefore, we plan to present this future work in the final version.

---

> > ### Comment · Reviewer_BWG1 · 2025-04-06
> >
> > I have read the comments by the authors and have no further questions.

---

> > > ### Author Response · Authors · 2025-04-06
> > >
> > > We sincerely appreciate your support and your insightful review comments. These comments will be carefully addressed in the revised version.

---

### Official Review · Reviewer_MM5X · 2025-03-14

**Overall Recommendation:** 2

**Summary:**

The paper "Towards Robustness and Explainability of Automatic Algorithm Selection" introduces a novel approach to algorithm selection using a directed acyclic graph (DAG) to model the causal relationships between problem features and algorithm features. The proposed method, DAG-based Algorithm Selection (DAG-AS), aims to enhance robustness against distribution shifts and improve explainability at both the model and instance levels. The paper demonstrates the effectiveness of DAG-AS through experiments on the ASlib benchmark, highlighting its advantages in terms of accuracy, robustness, and explainability.

**Claims And Evidence:**

The paper claims that DAG-AS improves robustness and explainability in algorithm selection tasks. The evidence provided includes experimental results showing that DAG-AS outperforms traditional methods in terms of PAR10 scores across multiple datasets. The paper also presents causal graphs and counterfactual explanations to support the claim of enhanced explainability.

**Essential References Not Discussed:**

N/A

**Experimental Designs Or Analyses:**

The experimental design includes performance comparisons, ablation studies, and robustness tests against distribution shifts. The authors use ten ASlib datasets with algorithm features and conduct experiments to evaluate the performance, robustness, and explainability of DAG-AS. The experiments are repeated multiple times to ensure reliability, and the results are presented in terms of PAR10 scores and causal graph analyses.

**Methods And Evaluation Criteria:**

The authors employ a causal learning framework using DAGs to model the relationships between problem and algorithm features. The evaluation criteria include the PAR10 score, which measures the performance of algorithm selection methods based on solution time and timeouts. The authors compare DAG-AS with several established methods and baselines, including ISAC, MCC, SATzilla11, SNNAP, SUNNY, VBS, and SBS.

**Other Comments Or Suggestions:**

The authors could provide more details on the computational complexity and scalability of the DAG-AS framework, especially for large-scale datasets.
It would be beneficial to explore the potential of DAG-AS in other domains beyond the ASlib benchmark to assess its generalizability.

**Other Strengths And Weaknesses:**

+
The paper introduces a novel approach to algorithm selection that leverages causal learning, which is a relatively unexplored area in this field.
The use of DAGs provides a clear and interpretable framework for understanding the relationships between problem and algorithm features.
The experimental results demonstrate the effectiveness of DAG-AS in improving robustness and explainability.

-
The paper's performance on certain datasets, such as GLUHACK-18 and SAT03-16-INDU, is not as strong as on others, indicating potential limitations in capturing causal relationships with limited training data.
The complexity of the causal learning framework may pose challenges for practical implementation and scalability.

**Questions For Authors:**

How does the computational complexity of DAG-AS compare to traditional algorithm selection methods, and what are the implications for scalability?
Can DAG-AS be applied to other domains or types of problems beyond those included in the ASlib benchmark?
How does the choice of problem and algorithm features impact the performance and explainability of DAG-AS?
Are there any specific types of distribution shifts or problem characteristics where DAG-AS may not perform well?

**Relation To Broader Scientific Literature:**

N/A

**Theoretical Claims:**

The paper theoretically claims that modeling the conditional distribution of algorithm features based on problem features using a DAG can improve robustness against distribution shifts and provide multi-level explainability. The authors support this claim with a theorem (Theorem 2.3) that establishes the feasibility of using causal models for algorithm selection tasks.

---

> ### Author Rebuttal · Authors · 2025-03-31
>
> > The paper's performance on GLUHACK-18 and SAT03-16-INDU, is not as strong as on others, indicating potential limitations ...
>
> We acknowledge that DAG-AS did not perform well on GLUHACK-18 and SAT03-16-INDU. However, it is unjustified to undermine the significance of DAG-AS merely based on its performance on these 2 datasets. Specifically,
> 1. It is challenging for any single method to achieve optimal performance across all problems. In this paper, we conducted comprehensive evaluation on 10 datasets. The results demonstrate that DAG-AS outperformed all the comparison methods on 8 of these data. Notably, on half of them, DAG-AS significantly outperformed other methods, which illustrates the distinct advantages of DAG-AS.
> 2. Regarding the DAG-AS's performance on GLUHACK-18 and SAT03-16-INDU, it may be due to the data violating the causal sufficiency assumption or the overly complex causal relations within the data. Nevertheless, these issues are not insurmountable in the causal learning field. There are numerous models specifically designed for complex causality in the causal learning domain, such as models for handling multivariate/non-linear causality, and models for scenarios where the causal sufficiency assumption is violated. The causal learning module of DAG-AS is a relatively simple MLP, mainly to highlight the impact of introducing causality into algorithm selection (AS). If further performance improvement is desired, more advanced causal learning models can be employed to replace Eq(8,9).
>
> > Computational complexity of DAG-AS
>
> There is no significant difference in computational complexity between DAG-AS and existing methods. Compared to traditional methods, DAG-AS needs to construct a causal DAG and reconstruct algorithm features, while traditional methods often directly predict the optimal algorithm. However, due to the sparsity of the DAG, the reconstruction model for each algorithm feature is actually small-scale. We have provided the running times comparison at the anonymous link: https://imgur.com/a/1w6eLIQ. It can be seen that the running time of DAG-AS is comparable to existing methods. Despite this, DAG-AS achieves remarkable performance gains, indicating its practicability.
>
> > Can DAG-AS be applied to other domains or types of problems beyond those included in the ASlib benchmark?
>
> DAG-AS is designed to be applicable to a wide range of AS scenarios, far beyond the scenarios in the ASlib benchmark. The key requirement for applying DAG-AS is the availability of the general configuration for the AS task, including problem features, algorithm features, and performance data. As long as these elements are provided, DAG-AS can be effectively utilized. Actually, the ASlib itself is also composed of problems from diverse domains.
>
> > How does the choice of problem and algorithm features impact the performance and explainability of DAG-AS?
>
> *Performance Impact*: Different features carry distinct information about the nature of the problem and algorithm. In datasets with a rich set of features that are informative and highly relevant to the algorithm's performance, DAG-AS can better capture the causal relationships. This is similar for other AS methods. Traditional algorithms aim for these features to be correlated with the algorithm's performance, while DAG-AS expects there to be causal relationships behind such correlations.
>
> *Explainability Impact*: When it comes to the impact on explainability, the nature of features matters depending on the user's goals. If users want to understand the decision-making logic of DAG-AS and interpret the causal relationships between features in the DAG, these features should have physical meaning, rather than being representations extracted by deep networks. However, if the sole expectation is for DAG-AS to make AS decisions, features without clear physical meaning can also be perfectly suitable. For example, since ASlib did not provide algorithm features for the BNSL-2016, we used a LLM to extract representations from the code of candidate algorithms. Despite the lack of obvious physical meaning in these features, DAG-AS still achieved the best performance.
>
> > Are there any specific types of distribution shifts or problem characteristics where DAG-AS may not perform well?
>
> The causal learning module in DAG-AS is essentially built on modeling conditional probabilities. Hence, it can mitigate the impact of covariate shift and prior probability shift. It has been verified in Fig.6 that DAG-AS can handle distribution shifts on problem features and optimal algorithm distributions well. However, DAG-AS may face challenges in dealing with shifts in $P(AF|PF)$. But in AS, $P(AF|PF)$ implies what characteristics an algorithm needs to solve a problem with specific feature values, which does not shift in a given AS task. As for problem characteristics, DAG-AS has no specific requirements regarding the problem domain, as mentioned in previous responses.

---

### Decision · Program_Chairs · 2025-05-01

**Decision:**

Accept (spotlight poster)

**Comment:**

The paper provides a well-founded contribution for explaining why certain algorithms perform well on certain instance sets using a DAG mechanism. There has not been much of a focus on explainability in optimization contexts, and especially not for algorithm selection. This paper is thus really pioneering methods in this area. The reviewers compliment both the theory and experimental results. The paper uses standard benchmarks in the area and draws some interesting conclusions. I note that the only negative review was flagged by the LLM detection software, and even ignoring the fact that it may have been written by an LLM, it is of low quality and offers no coherent argument for rejecting the paper.